# T2V-Turbo-v2: Enhancing Video Generation Model Post-Training through Data, Reward, and Conditional Guidance Design

**Jiachen Li**[1], **Qian Long**[2], **Jian Zheng**[3]*, **Xiaofeng Gao**[3]*, **Robinson Piramuthu**[3]*,
**Wenhu Chen**[4], **William Yang Wang**[1]
[1]UC Santa Barbara, [2]UC Los Angeles, [3]Amazon AGI, [4]University of Waterloo
[1]{jiachen_li, william}@cs.ucsb.edu, [2]longqian@ucla.edu,
[3]{nzhengji, gxiaofen, robinpir}@amazon.com [4]wenhuchen@uwaterloo.ca

**Project Page**: https://t2v-turbo-v2.github.io

**Videos: click to play in Adobe Acrobat**

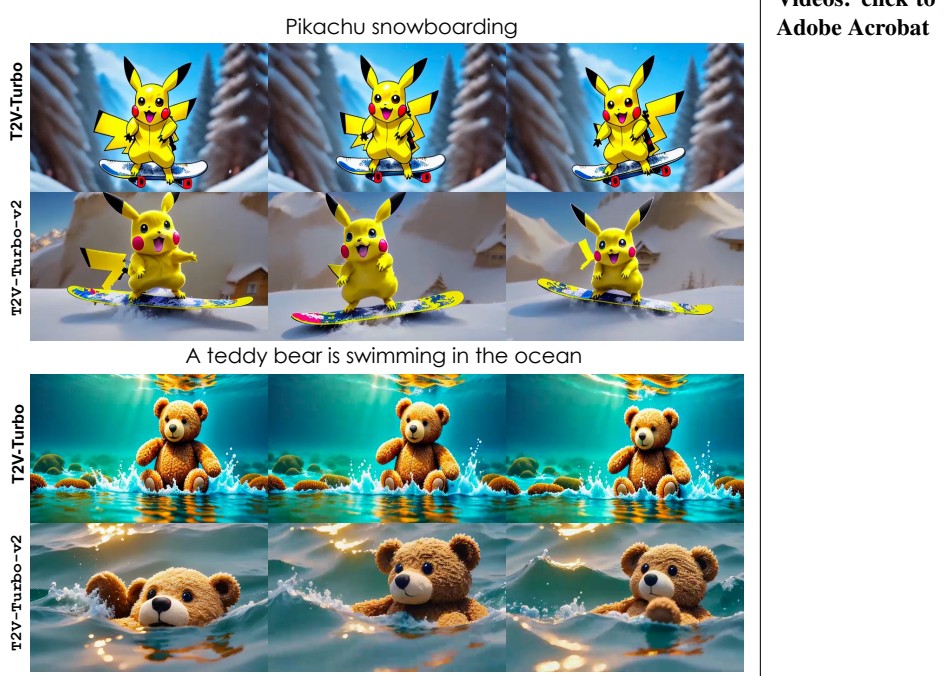

Figure 1: By meticulously designing and selecting the training datasets, reward models, and guidance, our `T2V-Turbo-v2` generates videos that are visually more appealing, semantically better aligned, and more dynamic compared to T2V-Turbo (Li et al., 2024b).

## ABSTRACT

In this paper, we focus on enhancing a diffusion-based text-to-video (T2V) model during the post-training phase by distilling a highly capable consistency model from a pretrained T2V model. Our proposed method, `T2V-Turbo-v2`, introduces a significant advancement by integrating various supervision signals, including high-quality training data, reward model feedback, and conditional guidance, into the consistency distillation process. Through comprehensive ablation studies, we highlight the crucial importance of tailoring datasets to specific learning objectives and the effectiveness of learning from diverse reward models for enhancing both the visual quality and text-video alignment. Additionally, we highlight the vast design space of conditional guidance strategies, which centers on designing an effective energy function to augment the teacher ODE solver. We demonstrate the potential of this approach by extracting motion guidance from

---

*This work does not relate to the author's position at Amazon.

the training datasets and incorporating it into the ODE solver, showcasing its effectiveness in improving the motion quality of the generated videos with the improved motion-related metrics from VBench and T2V-CompBench. Empirically, our `T2V-Turbo-v2` establishes a new state-of-the-art result on VBench, **with a Total score of 85.13**, surpassing proprietary systems such as Gen-3 and Kling.

# 1 INTRODUCTION

Diffusion-based (Sohl-Dickstein et al., 2015; Ho et al., 2020) neural video synthesis has been advancing at an unprecedented pace, giving rise to cutting-edge text-to-video (T2V) systems like Sora (Brooks et al., 2024), Kling (Kuaishou, 2024), DreamMachine (Luma, 2024) and Gen-3 (Gen-3, 2024). These models are capable of generating high-quality videos with detailed motion dynamics. However, the majority of these systems are proprietary, as pretraining them from scratch requires significant computational resources and access to extensive, human-curated video datasets, which are not readily accessible to the academic community. Consequently, a notable performance disparity has emerged between proprietary models and currently available open-source alternatives (Chen et al., 2024a; Guo et al., 2023; Wang et al., 2023a;c; Open-Sora, 2024; Lab & etc., 2024), which are trained on datasets with varying video quality, e.g., WebVid-10M (Bain et al., 2021).

Efforts have been directed toward bridging the performance gap. On the one hand, various video datasets (Tan et al., 2024; Nan et al., 2024; Chen et al., 2024b; Ju et al., 2024) with higher visual quality and detailed captions are released to the public domain. On the other hand, several methods have been proposed to enhance the sample quality of pretrained T2V models during the post-training phase. For example, InstructVideo (Yuan et al., 2023) and Vader (Prabhudesai et al., 2024) propose aligning pretrained T2V models by backpropagating gradients from image-text reward models (RMs). More recently, Li et al. (2024b) proposes T2V-Turbo, achieving fast and high-quality video generation by incorporating feedback from a mixture of RMs into the consistency distillation (CD) process (Song et al., 2023; Luo et al., 2023). However, these methods still employ the WebVid data to train the model and thus solely rely on the reward feedback to improve the generation quality. In addition, training-free methods (Feng et al., 2023; Mo et al., 2024; Xie et al., 2023) have emerged that seek to enhance generation quality by introducing conditional guidance during inference. For example, MotionClone (Ling et al., 2024) extracts the temporal attention matrix from a reference video as the motion prior and uses it to formulate an energy function that guides the T2V model's sampling process, achieving impressive results. However, this approach requires calculating the additional guidance term at each sampling step, imposing significant computational overhead, and requiring access to a high-quality reference video.

In this paper, we aim to advance the post-training of the T2V model by incorporating supervision signals from various sources. To this end, we introduce `T2V-Turbo-v2`, integrating various supervision signals from high-quality video datasets, reward feedback, and conditional guidance into the CD process. Additionally, we highlight the vast design space of conditional guidance strategy, which can be boiled down to designing an energy function to augment the teacher ODE solver together with the classifier-free guidance (CFG) (Ho & Salimans, 2021). In this paper, we empirically showcase the potential of this approach by leveraging the motion guidance from MotionClone to formulate our energy function. Regarding the reference video, our key insight is that, given a pair of video and caption, **the video itself naturally serves as the ideal reference when the generation prompt is its own caption**. Moreover, we remedy the substantial computational cost of calculating the conditional guidance by dedicating a data preprocessing phase before training the consistency model. Implementation-wise, we improve upon T2V-Turbo by eliminating the target network used to calculate the distillation target without experiencing training instability. The saved memory from the preprocessing phase and removing the target network allows us to perform full model training, whereas T2V-Turbo can only train its model with LoRA (Hu et al., 2021).

We design comprehensive experiments to investigate how different combinations of training datasets, RMs, and conditional guidance impact the performance of our `T2V-Turbo-v2`. Our design of RMs is more rigorous compared to T2V-Turbo, leveraging feedback from vision-language foundation models, e.g., CLIP (Radford et al., 2021) and the second stage model of InternVideo2 (InternV2) (Wang et al., 2024; 2023d), after observing that the baseline VideoLCM (Wang et al., 2023b) (VCM) fails to align the text and its generated on every dataset variants. We also empirically

show the benefits of learning from diverse RMs. Moreover, we identified that while training on datasets with higher visual quality and dense video captions benefits visual quality, the dense captions prevent the model from fully optimizing the benefits from the reward feedback, particularly due to the limited context length of existing RMs. To combat this issue, we carefully curate the training data for different learning objectives, leading to a substantial performance gain. These experiments **provide solid empirical evidence to motivate future research to develop long-context RMs**. Finally, to verify the motion quality improvement made by the incorporation of motion guidance, we leverage various metrics from both VBench (Huang et al., 2024) and T2V-CompBench (Sun et al., 2024) to access the motion quality of different methods.

We distill our `T2V-Turbo-v2` from VideoCrafter2 (Chen et al., 2024a). Empirically, we evaluate the performance of its 16-step generation on the VBench (Huang et al., 2024), both with and without augmenting the ODE solver with the motion guidance. Remarkably, both variants outperform all existing baselines on the VBench leaderboard. The variant that incorporates motion guidance sets a new SOTA on VBench, achieving a Total Score of 85.13, surpassing even proprietary systems such as Gen-3 (Gen-3, 2024) and Kling (Kuaishou, 2024).

Our contributions are threefold.

- A rigorous and thorough empirical investigation into the effects of training data, RMs, and conditional guidance design on the post-training phase of the T2V model, shedding light on future T2V post-training research.

- Establish a new SOTA Total Score on VBench, outperforming proprietary systems, including Gen-3 and Kling.

- Highlight the vast design space of energy functions to augment the ODE solver, demonstrating its potential by extracting motion priors from training videos to enhance T2V model training. To the best of our knowledge, this is the first work to introduce this approach.

## 2 PRELIMINARY

**Diffusion Sampling** The sampling process from a latent video diffusion model (Ho et al., 2022b) can be treated as solving an empirical probability flow ordinary differential equation (PF-ODE) (Song et al., 2020b). The sampling process executes in a reversed-time order, starts from a standard Gaussian noise $z_T$, and returns a clean latent $z_0$.

$$\mathrm{d}z_t = \left[ \boldsymbol{\mu}\left(t\right) z_t + \frac{1}{2}\sigma(t)^2 \boldsymbol{\epsilon}_\psi(z_t, \boldsymbol{c}, t) \right] \mathrm{d}t, \quad z_T \sim \mathcal{N}(\mathbf{0}, \mathbf{I}), \tag{1}$$

where $z_t \sim p_t(z_t)$ is the noisy latent at timestep $t$, $\boldsymbol{c}$ is the text prompt, and $\boldsymbol{\mu}(\cdot)$ and $\sigma(\cdot)$ are the drift and diffusion coefficients, respectively. The denoising model $\boldsymbol{\epsilon}_\psi(z_t, \boldsymbol{c}, t)$ is trained to approximate the score function $-\nabla \log p_t(z_t)$ via score matching, and is used to construct an ODE solver $\Psi$.

To improve the quality of conditional sampling, we can augment the denoising model $\boldsymbol{\epsilon}_\psi$ with classifier-free guidance (CFG) (Ho & Salimans, 2021) and the gradient of an energy function $\mathcal{G}$

$$\hat{\epsilon}_\psi(z_t, \boldsymbol{c}, t) = \epsilon_\psi\left(z_t, \boldsymbol{c}, t\right) + \omega\left(\epsilon_\psi\left(z_t, \boldsymbol{c}, t\right) - \epsilon_\psi\left(z_t, \varnothing, t\right)\right) + \lambda \nabla_{z_t}\mathcal{G}\left(z_t, t\right), \tag{2}$$

where $\omega$ and $\lambda$ are parameters controlling the guidance strength.

**Consistency Model (CM)** (Song et al., 2023; Luo et al., 2023) is proposed to accelerate the sampling from a PF-ODE. Specifically, it learns a consistency function $\boldsymbol{f} : (z_t, \boldsymbol{c}, t) \mapsto \boldsymbol{x}_\epsilon$ to directly map any $z_t$ on the PF-ODE trajectory to its origin, where $\epsilon$ is a fixed small positive number. We can model $\boldsymbol{f}$ with a CM $\boldsymbol{f}_\theta$ and distill it from a pretrained diffusion model, e.g., a denoising model $\boldsymbol{\epsilon}_\psi$, by minimizing the *consistency distillation* (CD) loss.

Consider discretizing the time horizon into $N - 1$ sub-intervals with $t_1 = \epsilon < t_2 < \ldots < t_N = T$, we can sample any $z_{t_{n+k}}$ ($k$ is the skipping interval of the ODE solver $\Psi$) from the PF-ODE trajectory and obtain its solution with $\Psi$. Conventional methods (Luo et al., 2023; Wang et al., 2023b; Li et al., 2024a;b) augment $\Psi$ with CFG and the solution is given as below

$$\hat{z}_{t_n}^{\Psi,\omega} \leftarrow z_{t_{n+k}} + (1+\omega)\Psi(z_{t_{n+k}}, t_{n+k}, t_n, \boldsymbol{c}; \psi) - \omega\Psi(z_{t_{n+k}}, t_{n+k}, t_n, \varnothing; \psi). \tag{3}$$

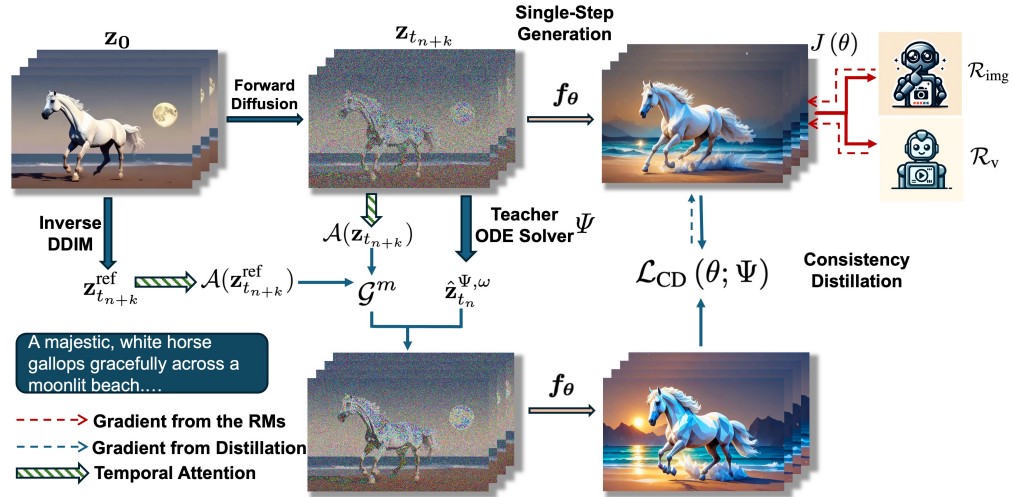

Figure 2: Training pipeline of our `T2V-Turbo-v2`. When augmenting the teacher PF-ODE solver $\Psi$ with CFG and motion guidance, we extract motion prior $\mathcal{A}(\boldsymbol{z}_{t_{n+k}}^{\text{ref}})$ from the training videos and distill it into the student CM $\boldsymbol{f}_\theta$ along with the CFG.

One can condition $\boldsymbol{f}_\theta$ on $\omega$, and formulate the CD loss by enforcing CM's *self-consistency* property:

$$L_{\text{CD}}\left(\theta, \theta^-; \Psi\right) = \mathbb{E}_{\boldsymbol{z}, \boldsymbol{c}, \omega, n}\left[d\left(\boldsymbol{f}_\theta\left(\boldsymbol{z}_{t_{n+k}}, \omega, \boldsymbol{c}, t_{n+k}\right), \boldsymbol{f}_{\theta^-}\left(\hat{\boldsymbol{z}}_{t_n}^{\Psi, \omega}, \omega, \boldsymbol{c}, t_n\right)\right)\right], \quad (4)$$

where $d(\cdot, \cdot)$ is a distance function. Conventional methods (Song et al., 2023; Luo et al., 2023; Wang et al., 2023b; Li et al., 2024b) update $\theta^-$ as the exponential moving average (EMA) of $\theta$, i.e., $\theta^- \leftarrow \texttt{stop\_grad}\left(\mu\theta + (1-\mu)\theta^-\right)$. In this paper, we find that $\theta^-$ is actually removable.

**Temporal Attention** is introduced to model the temporal dynamic of a video. Given a batch of video latents $\boldsymbol{z} \in \mathbb{R}^{B \times F \times C \times H \times W}$ with batch size $B$, $F$ frames, $C$ channels, and spatial dimensions $H \times W$, the temporal attention operator first reshapes the features by merging the spatial dimensions into the batch size dimension, leading to $\bar{\boldsymbol{z}} \in \mathbb{R}^{(B \times H \times W) \times F \times C}$. Then, it performs self-attention (Vaswani et al., 2017) along the frame axis to derive the temporal attention as below

$$\mathcal{A}(\boldsymbol{z}) = \texttt{Attention}\left(Q\left(\bar{\boldsymbol{z}}\right), K\left(\bar{\boldsymbol{z}}\right), V\left(\bar{\boldsymbol{z}}\right)\right) \in \mathbb{R}^{(B \times H \times W) \times F \times F}, \quad (5)$$

where $Q$, $K$, and $V$ are the Query, Key, and Value heads. And $\mathcal{A}(\boldsymbol{z})$ satisfies $\sum_{j=1}^F \mathcal{A}(\boldsymbol{z})_{(p,i,j)} = 1$.

**MotionClone** (Ling et al., 2024) represents the video motion as the temporal attention and enables the cloning of the motion from a reference video to control the video generation process. At its core, MotionClone augments the conventional classifier-free guidance (Ho & Salimans, 2021) with *Primary temporal-attention guidance* (PTA guidance) and *Location-aware semantic guidance* (LAS guidance). In this paper, we focus on the PTA guidance. Given a reference video, we first obtain its latent $\boldsymbol{z}_t^{\text{ref}}$ at the timestep $t$ via DDIM inversion (Song et al., 2020a). Then, we can derive the energy function $\mathcal{G}^m$ associated with PTA guidance at timestep $t$ as below

$$\mathcal{G}^m\left(\boldsymbol{z}_t, \boldsymbol{z}_t^{\text{ref}}, t\right) = \left\|\mathcal{M} \circ \left(\mathcal{A}(\boldsymbol{z}_t^{\text{ref}}) - \mathcal{A}(\boldsymbol{z}_t)\right)\right\|_2^2, \quad (6)$$

where $\circ$ denotes the Hadamard product and $\mathcal{M}$ is the temporal mask set to mask everything except for the highest activation along the temporal axis of $\mathcal{A}(\boldsymbol{z}_t^{\text{ref}})$, i.e.,

$$\mathcal{M}_{(p,i,j)} := \begin{cases} 1, & \text{if } \mathcal{A}(\boldsymbol{z}_t^{\text{ref}})_{(p,i,j)} = \max_k \mathcal{A}(\boldsymbol{z}_t^{\text{ref}})_{(p,i,k)} \\ 0, & \text{otherwise} \end{cases} \quad (7)$$

## 3 INTEGRATE REWARDS AND CONDITIONAL GUIDANCE INTO CONSISTENCY DISTILLATION

Conventional CD methods (Song et al., 2023; Luo et al., 2023; Li et al., 2024a;b) typically neglect the design space of conditional guidance methods and do not employ an energy function $\mathcal{G}$ to aug-

ment the ODE solver $\Psi$. With the conditional guidance from $\mathcal{G}$, we can derive the solution of the augmented ODE solver as $\hat{z}_{t_n}^{\Psi,\omega,\lambda} \leftarrow \hat{z}_{t_n}^{\Psi,\omega} + \lambda \nabla_{z_t} \mathcal{G}(z_t, t)$ with $\lambda$ controlling the conditional guidance strength. And similarly, we further condition our CM $f_\theta$ on $\lambda$, i.e., $f_\theta : (z_t, \omega, \lambda, c, t) \mapsto z_0$.

In this paper, we are particularly interested in leveraging the PTA guidance as our motion guidance $\mathcal{G}^m$. With CFG, the solution of the augmented solver can be given by

$$
\begin{aligned}
\hat{z}_{t_n}^{\Psi,\omega,\lambda} \leftarrow z_{t_{n+k}} &+ (1+\omega)\Psi(z_{t_{n+k}}, t_{n+k}, t_n, c; \psi) - \omega\Psi(z_{t_{n+k}}, t_{n+k}, t_n, \varnothing; \psi) \\
&+ \lambda \cdot \nabla_{z_{t_{n+k}}} \mathcal{G}^m \left( z_{t_{n+k}}, z_{t_{n+k}}^{\text{ref}}, t_{n+k} \right)
\end{aligned}, \tag{8}
$$

The core insight of our method is that given video datasets with decent motion quality, the training video itself naturally serves as the ideal reference when the generation prompt is its own caption. Therefore, we can always extract the motion information from the training video and employ it to guide the video generation. Following MotionClone (Ling et al., 2024), we only apply motion guidance to the first $\tau$ percent of the sampling steps, i.e., we explicitly set $\lambda = 0$ for $n < N(1 - \tau)$. Note that for $n \geq N(1 - \tau)$, we can still set $\lambda = 0$ for video without good motion quality.

## 3.1 Learning Objectives

Our preliminary experiment shows that the target network $f_{\theta^-}$ can be removed without affecting the training stability. This is empirically significant, as it frees us from maintaining the EMA $\theta^-$, saving a substantial amount of GPU memory. With our $f_\theta$ conditioned $\lambda$, our new CD loss can be derived by slightly modifying Eq. 4

$$
\mathcal{L}_{\text{CD}}(\theta; \Psi) = \mathbb{E}_{z,c,\omega,n} \left[ d \left( f_\theta \left( z_{t_{n+k}}, \omega, \lambda, c, t_{n+k} \right), \texttt{stop\_grad} \left( f_\theta \left( \hat{z}_{t_n}^{\Psi,\omega,\lambda}, \omega, \lambda, c, t_n \right) \right) \right) \right], \tag{9}
$$

Note that we do not distill the LAS guidance term from the original MotionClone, which has been demonstrated to enhance generation quality. Instead, we mitigate the potential quality loss by following T2V-Turbo (Li et al., 2024b), augmenting the conventional consistency distillation with an objective to maximize a mixture of RMs, including an image-text RM $\mathcal{R}_{\text{img}}$ and a video-text RM $\mathcal{R}_{\text{v}}$. The reward optimization objective $J$ can be formulated as below

$$
J(\theta) = \mathbb{E}_{z,c,n} \left[ \beta_{\text{img}} \sum_{m=1}^{M} \mathcal{R}_{\text{img}}(\hat{x}_0^m, c) + \beta_{\text{v}} \mathcal{R}_{\text{v}}(\hat{x}_0, c) \right], \quad \hat{x}_0 = \mathcal{D}\left( f_\theta \left( z_{t_{n+k}}, \omega, \lambda, c, t_{n+k} \right) \right), \tag{10}
$$

where $\beta_{\text{img}}$ and $\beta_{\text{v}}$ are the weighting parameters. Note that we can optimize multiple $\mathcal{R}_{\text{img}}$ and $\mathcal{R}_{\text{v}}$ with minimal change to Eq. 10. In Sec. 4.3, we investigate the effects of different choices of RMs. Our total training loss combines the CD loss and the reward optimization objective as follows:

$$
L(\theta; \Psi) = \mathcal{L}_{\text{CD}}(\theta; \Psi) - J(\theta). \tag{11}
$$

## 3.2 Data Processing and Training Procedures

It is important to note that calculating the gradient of an energy function $\mathcal{G}$ can be computationally expensive, consuming substantial memory. For example, using the motion guidance $\mathcal{G}^m$ as the energy function further requires performing DDIM inversion to obtain $z_{t_{n+k}}^{\text{ref}}$, which is too expensive to be done during each training iteration. For example, MotionClone's original implementation can consume over 40GB of GPU memory and require 3 minutes to perform the DDIM inversion.

Fortunately, we identify that the solution $\hat{z}_{t_n}^{\Psi,\omega,\lambda}$ can be pre-calculated before training the CM $f_\theta$. Appendix A describes the detailed procedures of our preprocessing procedures in Algorithm 1 and includes the pseudo-codes for training in Algorithm 2. Our training pipeline is depicted in Fig. 2.

## 4 Experimental Results

Our experiments aim to demonstrate our `T2V-Turbo-v2`'s ability to generate high-quality videos and unveil the key design choices contributing to our superior performance. Sec. 4.1 evaluate our method on VBench (Huang et al., 2024) from various dimensions against a broad array of baseline

Table 1: **Evaluation results on VBench** (Huang et al., 2024). We present the performance of our `T2V-Turbo-v2` with 16 inference steps, both with and without the application of Motion Guidance, and benchmark it against a wide range of baseline models. *Quality Score*, *Semantic Score*, and *Total Score* respectively reflect the visual quality, text-video alignment, and overall human preference of the generated videos. Table 11 in Appendix C provides a detailed breakdown for each evaluation dimension. The best result for each score is highlighted in bold, and the second-best result is underlined. Our `T2V-Turbo-v2` achieves SOTA results on VBench, outperforming all baseline methods, including proprietary systems such as Gen-3 and Kling, in terms of *Total Score*.

| | Pika | Gen-2 | Gen-3 | Kling | VideoCrafter2 | T2V-Turbo | | T2V-Turbo-v2 | |
| | | | | | | 4-step | 16-step | w/o MG | w/ MG |
|---|---|---|---|---|---|---|---|---|---|
| Quality Score | 82.92 | 82.47 | 84.11 | 83.39 | 82.20 | 82.57 | 82.27 | 84.08 | **85.13** |
| Semantic Score | 71.77 | 73.03 | 75.17 | 75.68 | 73.42 | 74.76 | 73.43 | **78.33** | 77.12 |
| Total Score | 80.69 | 80.58 | 82.32 | 81.85 | 80.44 | 81.01 | 80.51 | 82.93 | **83.52** |

methods. We then perform thorough ablation studies to demonstrate the importance of carefully selecting training datasets (Sec. 4.2), reward models (Sec. 4.3), and guidance methods (Sec. 4.4).

**Settings**. We distill our `T2V-Turbo-v2` from VideoCrafter2 (Chen et al., 2024a). All our models are trained on 8 NVIDIA A100 GPUs for 8K gradient steps without gradient accumulation. We use a batch size of 3 to calculate the CD loss and 1 to optimize the reward objective on each GPU device. During optimization of the image-text reward model $\mathcal{R}_{img}$, we randomly sample 2 frames from each video by setting $M = 2$. The learning rate is set to $1e-5$, and the guidance scale is defined within the range $[\omega_{min}, \omega_{max}] = [5, 15]$. We use DDIM (Song et al., 2020a) as our ODE solver $\Psi$, with a skipping step parameter of $k = 5$. For motion guidance (MG), we set the motion guidance percentage $\tau = 0.5$ and strength $\lambda = 500$.

## 4.1 COMPARISON WITH SOTA METHODS ON VBENCH

We train two variants of our `T2V-Turbo-v2`. Specifically, `T2V-Turbo-v2` w/o MG is trained using the CFG-augmented solver (Eq. 3) without motion guidance, whereas `T2V-Turbo-v2` w/ MG includes motion guidance by using solver in Eq. 8. We train on a mixed dataset VG + WV, which consists of equal portions of VidGen-1M (Tan et al., 2024) and WebVid-10M (Bain et al., 2021). While the CD loss is optimized across the entire dataset, the reward objective Eq. 10 is optimized using only WebVid data. We utilize a combination of HPSv2.1 (Wu et al., 2023a) and ClipScore (Radford et al., 2021) as our $\mathcal{R}_{img}$, applying the same weight of $\beta_{img} = 0.2$. Additionally, we employ the second-stage model of InternVideo2 (InternV2) (Wang et al., 2024) as the our $\mathcal{R}_v$ with $\beta_v = 0.5$. The rationale behind these design choices is elaborated in the ablation sections.

Table 1 compares the 16-step generation of our methods with selective baselines from the VBench leaderboard[1], including Gen-2 (Esser et al., 2023), Gen-3 (Esser et al., 2023), Pika (Pika Labs, 2023), VideoCrafter1 (Chen et al., 2023), VideoCrafter2 (Chen et al., 2024a), Kling (Kuaishou, 2024), and the 4-step and 16-step generations of T2V-Turbo (Li et al., 2024b). Except for the 16-step generations from T2V-Turbo, the performance of the other baseline method is directly reported from the VBench leaderboard. To evaluate the 16-step generation of our method and T2V-Turbo, we carefully follow VBench's evaluation protocols by generating 5 videos for each prompt. The *Quality Score* assesses the visual quality of the generated videos across 7 dimensions, while the *Semantic Score* measures the alignment between text prompts and generated videos across 9 dimensions. The *Total Score* is a weighted sum of the *Quality Score* and *Semantic Score*. Appendix B provides further details, including explanations for each dimension of VBench.

Both variants of our `T2V-Turbo-v2` consistently surpass all baseline methods on VBench in terms of Total Score, outperforming even proprietary systems such as Gen-3 and Kling. **Our models establish a SOTA on VBench as of the submission date**. The superior performance of our `T2V-Turbo-v2` without motion guidance (w/o MG) compared to T2V-Turbo underscores the importance of carefully selecting training datasets and reward models (RMs). While the 16-step results of T2V-Turbo underperform its 4-step counterpart, Appendix C.1 shows that our `T2V-Turbo-v2` effectively scales with increased inference steps and still outperforms T2V-Turbo with 4 steps.

---

[1]`https://huggingface.co/spaces/Vchitect/VBench_Leaderboard`

Table 2: Ablation studies on the design of training datasets. While VCM performed best on OV, `T2V-Turbo-v2` w/o MG only achieves modest improvements on OV but excels on VG+WV, highlighting the importance of curating specialized datasets to fully enhance model performance.

| | VCM | | | | | T2V-Turbo-v2 w/o MG | | | | |
|---|---|---|---|---|---|---|---|---|---|---|
| | OV | VG | WV | OV + WV | VG + WV | OV | VG | WV | OV + WV | VG + WV |
| Quality Score | **83.62** | 82.24 | 81.31 | 80.00 | 82.95 | 84.04 | 82.28 | 83.41 | 82.32 | **84.08** |
| Semantic Score | **61.93** | 58.06 | 55.51 | 46.53 | 60.65 | 68.73 | 72.22 | 73.04 | 75.74 | **78.33** |
| Total Score | **78.52** | 77.41 | 76.15 | 73.30 | 78.49 | 80.97 | 80.26 | 81.34 | 81.00 | **82.93** |

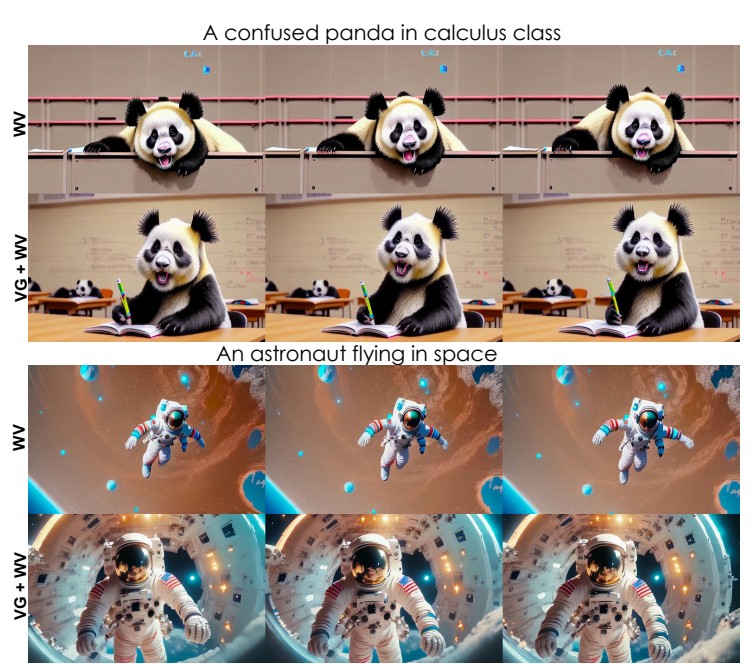

Figure 3: Trained on the VG + WV data, which are tailored for specific learning objectives, our `T2V-Turbo-v2` generates videos with improved visual quality and enhanced text-video alignment compared to training solely on the WV data of varying quality. **Play videos in Adobe Acrobat**.

## 4.2 ABLATION STUDIES ON THE DESIGN OF TRAINING DATASETS

To demonstrate the importance of training data selection, we experiment with VidGen-1M (Tan et al., 2024) (VG), OpenVid-1M (Nan et al., 2024) (OV), WebVid-10M (Bain et al., 2021) (WV), and their combinations. Specifically, OV and VG contain videos with high visual quality and detailed captions, whereas the conventional WV contains videos with mixed quality and short captions. *VG + WV* combines equal portions of VG and WV. *OV + WV* combines equal portions of OV and WV. We train VCM and our `T2V-Turbo-v2` w/o MG using the same set of RMs as in Sec. 4.1 for each dataset and report the evaluation results in Table 2 (per-dimension scores are in Table 8).

Surprisingly, `T2V-Turbo-v2`'s **performance does not scale in parallel with VCM's performance**. While VCM attains its highest performance on OV, incorporating reward feedback only leads to modest performance gain: a gain of 0.41 (83.63 → 84.04) on Quality Score, 5.80 (61.9 → 68.73) on Semantic Score, and 2.45 (78.52 → 80.97) on Total Score. A similar phenomenon is observed with VG. In contrast, VCM's performance on WV is relatively low, but integrating reward feedback yields substantial gain, boosting Quality Score by 2.10 (81.31 → 83.41), Semantic Score by 17.53 (55.5 → 73.04), and Total Score by 5.19 (76.15 → 81.34).

Based on these results, we hypothesize that the modest performance gains on the higher-quality datasets, OV and VG, are due to their excessively long video captions, which exceed the maximum context length of our RMs. For instance, the maximum context length of HPSv2.1 and CLIP is 77 tokens, while InternV2 has a maximum context length of only 40 tokens. As a result, these RMs can

Table 3: Ablation studies on the design of reward models. We train our `T2V-Turbo-v2` w/ MG on VG + WV with different combinations of RMs. While HPSv2.1 contributes the most to T2V-Turbo's performance as reported by Li et al. (2024b), incorporating feedback from a diverse set of RMs is crucial for our good performance, highlighting that RM selection is dataset dependent.

| | - | HPSv2.1 | CLIP | InternV2 | HPSv2.1 + CLIP | HPSv2.1 + InternV2 | CLIP + InternV2 | HPSv2.1 + CLIP + InternV2 |
|---|---|---|---|---|---|---|---|---|
| Quality Score | 82.78 | 82.76 | 83.11 | 83.02 | 82.13 | 84.17 | 84.05 | **85.13** |
| Semantic Score | 64.01 | 64.28 | 70.80 | 74.75 | **77.66** | 73.40 | 74.01 | 77.12 |
| Total Score | 79.02 | 79.07 | 80.65 | 81.37 | 81.24 | 82.02 | 82.04 | **83.52** |

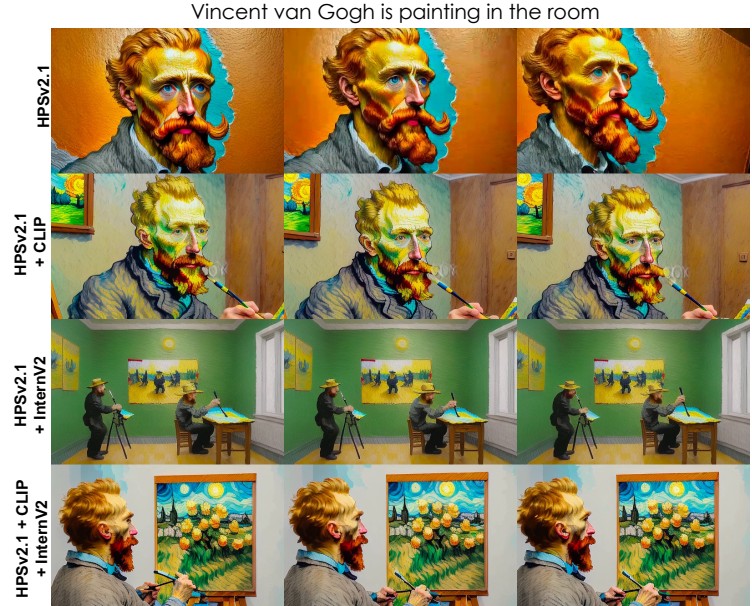

Figure 4: Qualitative comparison when learning with different combinations of RMs. Incorporating feedback from HPSv2.1 is not enough to achieve satisfactory text-video alignment. Learning from diverse RMs enhances both visual and semantic quality. **Play videos in Adobe Acrobat**.

only operate optimally when trained on datasets with shorter captions. To fully leverage high-quality training videos and maximize the impact of reward feedback, a plausible approach would be using visually appealing data for CD loss and short-captioned data for reward optimization. However, as demonstrated in Appendix C.2, this decoupling can result in undesired color distortion in the generated videos, potentially due to the substantial domain shift between the two datasets regarding the prompt space. This finding suggests that **the CD loss acts as an essential regularizer that prevents reward over-optimization**. Hence, we minimize CD loss using entire datasets while restricting reward optimization to short-captioned datasets. In Appendix C.2, we empirically demonstrate that this approach yields superior results compared to using the entire dataset for reward optimization.

As shown in Table 2, `T2V-Turbo-v2` achieves the highest scores on the VG + WV datasets. Fig. 3 compares videos generated from the model trained on VG + WV and the model trained on WV. However, while `T2V-Turbo-v2` shows notable improvements over VCM on OV + WV, its overall performance remains moderate, possibly due to the substantial domain gap between OV and WV.

## 4.3 ABLATION STUDIES ON THE DESIGN OF REWARD MODELS

From Table 2, we can conclude that **VCM falls short in aligning text and video** from the results. None of the VCM variants achieve a satisfactory Semantic Score, which motivates us to incorporate vision-language foundation models (Bommasani et al., 2021), such as CLIP and InternV2, to enhance text-video alignment in addition to feedback from HPSv2.1. In this section, we perform comprehensive ablation studies to assess the feedback from each model by training `T2V-Turbo-v2` w/ MG with different combinations of HPSv2.1, CLIPScore, and InternV2 on VG + MV.

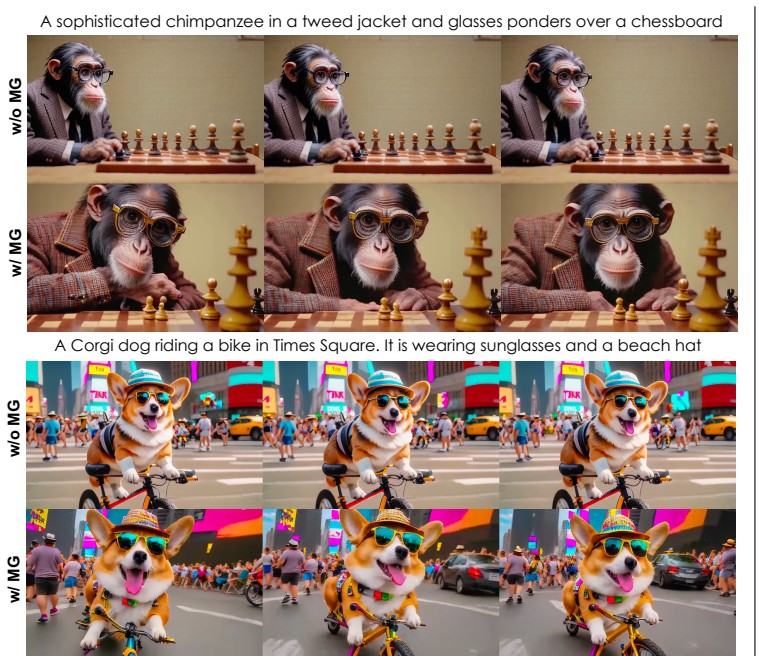

Figure 5: Comparison between the generations from `T2V-Turbo-v2` w/o MG (Top) and w/ MG (Bottom). Integrating the Motion guidance leads to richer video motion that aligns with the prompt.

Table 4: Ablation studies on the effectiveness of the motion guidance. Augmenting the PF-ODE solver with motion guidance improves performance in the *Human Action*, *Dynamic Degree*, *Motion Binding*, *Action Binding* and *Dynamic Attribute* without decreasing *Motion Smoothness*.

| | Human Action | Dynamic Degree | Motion Smooth. | Motion Binding | Action Binding | Dynamic Attr. |
|---|---|---|---|---|---|---|
| `T2V-Turbo-v2` | | | | | | |
| w/o MG | 97.17 | 61.39 | 97.00 | 22.15 | 72.73 | 20.05 |
| w/ MG | **97.35** | **90.00** | **97.07** | **24.38** | **74.60** | **20.46** |

As illustrated in Table 3 (per-dimension scores are in Table 9) and Fig. 4, learning from a more diverse set of RMs results in better performance. Relying solely on feedback from HPSv2.1 results in only minimal enhancements in video quality compared to the baseline VCM trained without reward feedback. This contrasts with the findings of Li et al. (2024b), where HPSv2.1 significantly contributed to T2V-Turbo's performance when purely trained on the WV datasets. This discrepancy underscores that the impact of reward feedback is highly dependent on the dataset, emphasizing the importance of carefully designing the RM sets to achieve optimal performance.

## 4.4 ABLATION STUDIES ON THE EFFECT OF MOTION GUIDANCE

Table 1 demonstrates that augmenting $\Psi$ with motion guidance, $\mathcal{G}_t$, improves performance on VBench. To further evaluate the improvements in terms of video motion quality, we provide scores for *Human Action*, *Dynamic Degree*, and *Motion Smoothness* across different methods in Table 4. For a more comprehensive assessment of motion quality, we further report the *Motion Binding*, *Action Binding* and *Dynamic Atribute* scores from T2V-CompBench (Sun et al., 2024). Experimental results indicate that incorporating motion guidance enhances performance and improves all metrics listed in Table 4. Fig. 5 further includes a qualitative comparison between different the two variants. Additional qualitative results are included in Fig. 7 and Fig. 8.

## 5 RELATED WORK

**Diffusion-based T2V Models**. Conventional T2V models often rely on large-scale image datasets for training (Ho et al., 2022b; Wang et al., 2023a; Chen et al., 2023; 2024a; Ho et al., 2022a) or adopt weights from pre-trained text-to-image (T2I) models (Zhang et al., 2023; Blattmann et al.,

2023; Khachatryan et al., 2023). For example, LaVie (Wang et al., 2023c) begins training with WebVid-10M and LAION-5B before fine-tuning on a curated internal dataset of 23 million videos. Text-image datasets, such as LAION (Schuhmann et al., 2022), tend to be more than ten times larger than open-source video-text datasets like WebVid-10M (Bain et al., 2021), offering higher spatial resolution and greater diversity (Wang et al., 2023a). Recently, high-quality video datasets (Tan et al., 2024; Nan et al., 2024; Chen et al., 2024b; Wang et al., 2023d; Yang et al., 2024) with dense captions are collected and released to the public. In this paper, we aim to leverage the supervision signals from high-quality video to improve a pretrained T2V model during the post-training phase.

**Vision-and-language Reward Models**. Several open-source image-text reward models (RMs) have been developed to mirror human preferences for a given image-text pair, including HPS (Wu et al., 2023b;a), ImageReward (Xu et al., 2024), and PickScore (Kirstain et al., 2024). These models are typically fine-tuned from image-text foundation models like CLIP (Radford et al., 2021) and BLIP (Li et al., 2022) with human preference data. Recently, VideoScore (He et al., 2024) is released to reflect human preference on video-text pair. In this paper, we choose our RMs based on the fact that VCM struggles to achieve satisfactory text-video alignment on all of our training dataset variants. Thus, we leverage vision-language foundation models CLIP and InternV2, along with HPSv2.1, to improve the semantic quality of the generated videos. We perform thorough experiments to show that our model benefits from a diverse set of RMs.

**Learning from Human/AI Feedback** is an effective method for aligning generative models with human preferences (Leike et al., 2018; Ziegler et al., 2019; Ouyang et al., 2022; Stiennon et al., 2020; Rafailov et al., 2024). In the domain of image generation, various approaches have been introduced to align T2I models with human preferences, including reinforcement learning (RL)-based methods (Fan et al., 2024; Prabhudesai et al., 2023; Zhang et al., 2024) and backpropagation-based reward fine-tuning techniques (Clark et al., 2023; Xu et al., 2024; Prabhudesai et al., 2023). Recently, InstructVideo (Yuan et al., 2023) and Vader (Prabhudesai et al., 2024) extended reward fine-tuning to optimize T2V models. Our method extends T2V-Turbo (Li et al., 2024b), integrating supervision from high-quality datasets, diverse reward feedback, and conditional guidance.

**Training-Free Conditional Guidance** has been widely adopted in controlling image generations (Feng et al., 2023; Mo et al., 2024; Cao et al., 2023; Epstein et al., 2023; Ge et al., 2023; Patashnik et al., 2023; Xie et al., 2023; Zhao et al., 2022) and achieves great success. Motion-Clone (Ling et al., 2024) tackles T2V generation by leveraging temporal attention from a reference video to guide the generation process. In this paper, we leverage the same motion guidance from training videos to augment the ODE solver, enhancing motion quality of the generated videos.

## 6 CONCLUSION AND LIMITATIONS

In this paper, we present `T2V-Turbo-v2`, which integrates additional supervision signals from high-quality data, diverse reward feedback, and conditional guidance into the consistency distillation process. Notably, the 16-step generations from `T2V-Turbo-v2` establish a new state-of-the-art result on VBench, surpassing both its teacher VideoCrafter2 and proprietary systems such as Gen-3 and Kling. Through comprehensive ablation studies, we demonstrate the critical importance of tailoring data to specific training objectives. Additionally, we observe that VCM, without reward feedback, struggles to align text with the generated videos, highlighting the need to align with vision-language foundation models to improve text-video alignment. Furthermore, we identify and explore the vast design space of energy function, which can be used to augment the teacher ODE solver. We empirically validate the potential of this approach by showing that incorporating motion priors extracted from training data enhances the motion quality of the generated videos.

While our `T2V-Turbo-v2` demonstrates impressive empirical results, it is important to acknowledge certain limitations. One key constraint is that our approach cannot fully capitalize on high-quality datasets, such as OpenVid-1M, due to the limited context length of existing reward models. Additionally, the teacher model used in this work, VideoCrafter2, also relies on the CLIP text encoder for processing the generation text prompt, which may limit its ability to serve as the teacher ODE solver to handle dense video captions. Therefore, developing long-context reward models would benefit future post-training research in video generation models. Furthermore, future T2V models should incorporate text encoders capable of comprehending longer and more detailed prompts, thereby enhancing their capacity to generate richer and more aligned video content.

## ETHICS STATEMENT

In conducting this research, we have adhered to ethical guidelines to ensure integrity and transparency throughout the study. Our work does not involve human subjects, and therefore, no IRB approval was required. The datasets and models used in this study, such as OpenVid-1M, VigGen-1M, WebVid-10M, and VideoCrafter2, are publicly available and adhere to relevant data usage policies. We have taken care to use datasets that respect individual privacy, and no personally identifiable information has been included in our research.

Since we plan to release the code and models in the future, we are committed to formulating detailed guidelines to prevent the misuse of our models. These guidelines will emphasize ethical usage, discourage harmful applications, and promote responsible AI practices. Additionally, we will explore and incorporate safeguard mechanisms into our generation pipeline to mitigate risks associated with misuse, ensuring that the released models and code are aligned with ethical standards and safety considerations. This proactive approach is part of our ongoing responsibility to contribute to the safe and positive development of AI technologies.

## REPRODUCIBILITY STATEMENT

Our experiments are conducted with all open-sourced codes and training data. Our implementation codes have been included in the supplementary material and will be released to the public in a GitHub repository without breaking the double-blind rules.

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

# A    PSEUDO-CODES OF OUR T2V-TURBO-v2'S DATA PREPROCESSING AND TRAINING PIPELINE

Algorithm 1 and Algorithm 2 presents the pseudo-codes for data preprocessing and training, respectively.

---

**Algorithm 1** Data Preprocessing Pipeline

---

**Require:** text-video dataset $\mathcal{D}$, ODE solver $\Psi$, noise schedule $\alpha(t), \beta(t)$, guidance scale interval $[\omega_{\min}, \omega_{\max}]$, skipping interval $k$, guidance percentage $\tau$, VAE encoder $\mathcal{E}$.
  Initialize the processed dataset: $\mathcal{D}_z \leftarrow \{\}$
  **for** $(\boldsymbol{x}, \boldsymbol{c}) \in \mathcal{D}$ **do**
    Encode the video $\boldsymbol{x}$ into latent space: $\boldsymbol{z} = E(\boldsymbol{x})$
    Sample the time index: $n \sim \mathcal{U}[1, N - k], \omega \sim [\omega_{\min}, \omega_{\max}]$
    # Obtain the solution with the PF-ODE solver
    Sample $\boldsymbol{z}_{t_{n+k}} \sim \mathcal{N}\left(\alpha\left(t_{n+k}\right)\boldsymbol{z}; \sigma^2\left(t_{n+k}\right)\mathbf{I}\right)$
    $\hat{\boldsymbol{z}}_{t_n}^{\Psi,\omega} \leftarrow \boldsymbol{z}_{t_{n+k}} + (1+\omega)\Psi(\boldsymbol{z}_{t_{n+k}}, t_{n+k}, t_n, \boldsymbol{c}; \psi) - \omega\Psi(\boldsymbol{z}_{t_{n+k}}, t_{n+k}, t_n, \varnothing; \psi)$
    # Obtain the motion guidance
    **if** $n > N \cdot (1 - \tau)/k$ **then**
      Obtain the temporal attention $\mathcal{A}(\boldsymbol{z}_{t_{n+k}}; \psi)$
      Perform DDIM inversion to obtain $\boldsymbol{z}_{t_{n+k}}^{\text{ref}}$
      Obtain the temporal attention $\mathcal{A}(\boldsymbol{z}_{t_{n+k}}^{\text{ref}}; \psi)$
      Calculate attention mask $\mathcal{M}$ by following Eq. 7
      Calculate the motion guidance: $\mathcal{G} = \nabla_{\boldsymbol{z}_{t_{n+k}}}\left\|\mathcal{M} \circ \left(\mathcal{A}(\boldsymbol{z}_{t_{n+k}}^{\text{ref}}) - \mathcal{A}(\boldsymbol{z}_{t_{n+k}})\right)\right\|_2^2$
    **else**
      $\mathcal{G} = \boldsymbol{0}$
    **end if**
    Update $\mathcal{D}_z \leftarrow \mathcal{D}_z \cup \{\boldsymbol{z}_{t_{n+k}}, \hat{\boldsymbol{z}}_{t_n}^{\Psi,\omega}, \mathcal{G}, \omega, \boldsymbol{c}\}$
  **end for**
  **return** $\mathcal{D}_z$

---

**Algorithm 2** T2V-Turbo-v2 Training Pipeline

---

**Require:** processed latent dataset $\mathcal{D}_z$, initial model parameter $\theta$, learning rate $\eta$, distance metric $d$, decoder $\mathcal{D}$, image-text RM $\mathcal{R}_{\text{img}}$, video-text RM $\mathcal{R}_{\text{v}}$, reward scale $\beta_{\text{img}}$ and $\beta_{\text{v}}$, motion guidance scale $\lambda$.
  **repeat**
    Sample $(\boldsymbol{z}_{t_{n+k}}, \hat{\boldsymbol{z}}_{t_n}^{\Psi,\omega}, \mathcal{G}, \omega, \boldsymbol{c}) \sim \mathcal{D}_z$
    $\hat{\boldsymbol{z}}_{t_n}^{\Psi,\omega,\lambda} \leftarrow \hat{\boldsymbol{z}}_{t_n}^{\Psi,\omega} + \lambda \cdot \mathcal{G}$
    $\hat{\boldsymbol{x}}_0 = \mathcal{D}\left(\boldsymbol{f}_\theta\left(\boldsymbol{z}_{t_{n+k}}, \omega, \boldsymbol{c}, t_{n+k}\right)\right)$
    $J_{\text{img}}(\theta) = \mathbb{E}_{\hat{\boldsymbol{x}}_0, \boldsymbol{c}}\left[\sum_{m=1}^{M} \mathcal{R}_{\text{img}}\left(\hat{\boldsymbol{x}}_0^m, \boldsymbol{c}\right)\right]$
    $J_{\text{vid}}(\theta) = \mathbb{E}_{\hat{\boldsymbol{x}}_0, \boldsymbol{c}}\left[\mathcal{R}_{\text{v}}\left(\hat{\boldsymbol{x}}_0, \boldsymbol{c}\right)\right]$
    $L_{\text{CD}} = d\left(\boldsymbol{f}_\theta\left(\boldsymbol{z}_{t_{n+k}}, \omega, \boldsymbol{c}, t_{n+k}\right), \texttt{stop\_grad}\left(\boldsymbol{f}_\theta\left(\hat{\boldsymbol{z}}_{t_n}^{\Psi,\omega}, \omega, \boldsymbol{c}, t_n\right)\right)\right)$
    $\mathcal{L}\left(\theta; \Psi\right) \leftarrow L_{\text{CD}} - \beta_{\text{img}} J_{\text{img}}(\theta) - \beta_{\text{vid}} J_{\text{vid}}(\theta)$
    $\theta \leftarrow \theta - \eta\nabla_\theta\mathcal{L}\left(\theta; \Psi\right)$
  **until** convergence

---

## B   FURTHER DETAILS ABOUT VBENCH

In this section, we provide an overview of the metrics used in VBench (Huang et al., 2024), followed by a discussion of how the **Quality Score**, **Semantic Score**, and **Total Score** are derived. For further details, we encourage readers to consult the original VBench paper.

The **Quality Score** is determined using the following metrics:

- **Subject Consistency** (Subject Consist.): This metric measures the similarity of DINO (Caron et al., 2021) features across frames.
- **Background Consistency** (BG Consist.): This is calculated based on the CLIP (Radford et al., 2021) feature similarity across frames.
- **Temporal Flickering** (Temporal Flicker.): The mean absolute difference between frames is used to quantify flickering.
- **Motion Smoothness** (Motion Smooth.): Motion priors from the video frame interpolation model (Li et al., 2023b) assess the smoothness of motion.
- **Aesthetic Quality**: This metric is based on the average of aesthetic scores generated by the LAION aesthetic predictor (Schuhmann et al., 2022).
- **Dynamic Degree**: The RAFT model (Teed & Deng, 2020) calculates the dynamic level of the video.
- **Imaging Quality**: The MUSIQ (Ke et al., 2021) predictor is used to obtain the results.

The **Quality Score** is the weighted sum of these normalized metrics, with all metrics assigned a weight of 1, except for **Dynamic Degree**, which is weighted at 0.5.

The following metrics contribute to the **Semantic Score**:

- **Object Class**: The success rate of generating the intended object is assessed using GRiT (Wu et al., 2022).
- **Multiple Object**: GRiT (Wu et al., 2022) also evaluates how well multiple objects are generated as specified by the prompt.
- **Human Action**: UMT (Li et al., 2023a) is used to assess the depiction of human actions.
- **Color**: This metric checks if the color in the output matches the expected color using GRiT (Wu et al., 2022).
- **Spatial Relationship** (Spatial Relation.): Calculated via a rule-based method similar to (Huang et al., 2023a).
- **Scene**: The video caption generated by Tag2Text (Huang et al., 2023b) is compared to the scene described in the prompt.
- **Appearance Style** (Appear Style.): This metric uses ViCLIP (Wang et al., 2023d) to match the video's appearance style to the prompt's style description.
- **Temporal Style**: The similarity between the video feature and the temporal style description from ViCLIP (Wang et al., 2023d) is evaluated.
- **Overall Consistency** (Overall Consist.): The overall alignment between the video feature and the full text prompt is measured using ViCLIP (Wang et al., 2023d).

The **Semantic Score** is the mean of the normalized values of the above metrics. Finally, the **Total Score** is computed by taking the weighted sum of the **Quality Score** and the **Semantic Score**, using the formula:

$$\text{Total Score} = \frac{4 \cdot \text{Quality Score} + \text{Semantic Score}}{5} \tag{12}$$

# C  ADDITIONAL ABLATION RESULTS

## C.1  ABLATION STUDIES ON THE NUMBER OF INFERENCE STEPS.

We investigate the impact of varying the number of inference steps in Table 5. In general, increasing the number of inference steps leads to improved visual quality and better text-video alignment for our `T2V-Turbo-v2`. Our inference is performed using the BFloat16 data type. The 4-step sampling takes approximately 1 second, the 8-step sampling takes around 1.5 seconds, and the 16-step sampling takes about 3 seconds.

Table 5: Ablation studies on the number of inference steps. We collect the 4-step, 8-step, and 16-step generation from our `T2V-Turbo-v2` and compare their performance on VBench.

| Steps | Total Score | Quality Score | Subject Consist. | BG Consist. | Temporal Flickering | Motion Smooth. | Aesthetic Quality | Dynamic Degree | Image Quality |
|---|---|---|---|---|---|---|---|---|---|
| 4 | 82.34 | 83.93 | 94.30 | 94.80 | 96.82 | 97.17 | 61.52 | 84.72 | 72.77 |
| 8 | 83.05 | 84.74 | 95.03 | 95.86 | 97.23 | 97.14 | 62.30 | 88.06 | 72.32 |
| 16 | 83.52 | 85.13 | 95.50 | 96.71 | 97.35 | 97.07 | 62.61 | 90.00 | 71.78 |

| Steps | Semantic Score | Object Class | Multiple Objects | Human Action | Color | Spatial Relation. | Scene | Appear. Style | Temporal Style | Overall Consist. |
|---|---|---|---|---|---|---|---|---|---|---|
| 4 | 75.97 | 95.57 | 54.91 | 96.80 | 94.04 | 38.56 | 52.69 | 24.76 | 27.09 | 28.65 |
| 8 | 76.31 | 96.34 | 57.36 | 96.00 | 92.83 | 42.86 | 52.51 | 24.36 | 27.04 | 28.36 |
| 16 | 77.12 | 95.33 | 61.49 | 96.20 | 92.53 | 43.32 | 56.40 | 24.17 | 27.06 | 28.26 |

Note that the results of T2V-Turbo (Li et al., 2024b) reported in VBench (Huang et al., 2024) leaderboard is obtained with 4 function evaluation steps. To ensure a fair comparison between our `T2V-Turbo-v2` and T2V-Turbo, we also evaluate the 4-step generation of our `T2V-Turbo-v2` and compare it with T2V-Turbo in Table 6. Our `T2V-Turbo-v2` still outperforms T2V-Turbo in terms of Quality Score, Semantic Score, and Total Score.

Table 6: Comparison of 4-step generation between `T2V-Turbo-v2` and T2V-Turbo. Our `T2V-Turbo-v2` outperforms T2V-Turbo in Quality Score, Semantic Score, and Total Score.

| Models | Total Score | Quality Score | Subject Consist. | BG Consist. | Temporal Flicker. | Motion Smooth. | Aesthetic Quality | Dynamic Degree | Image Quality |
|---|---|---|---|---|---|---|---|---|---|
| T2V-Turbo | 81.01 | 82.57 | **96.28** | **97.02** | **97.48** | **97.34** | **63.04** | 49.17 | 72.49 |
| T2V-Turbo-v2 | **82.34** | **83.93** | 94.30 | 94.80 | 96.82 | 97.17 | 61.52 | **84.72** | **72.77** |

| Models | Semantic Score | Object Class | Multiple Objects | Human Action | Color | Spatial Relation. | Scene | Appear. Style | Temporal Style | Overall Consist. |
|---|---|---|---|---|---|---|---|---|---|---|
| T2V-Turbo | 74.76 | 93.96 | 54.65 | 95.20 | 89.90 | **38.67** | **55.58** | 24.42 | 25.51 | 28.16 |
| T2V-Turbo-v2 | **75.97** | **95.57** | **54.91** | **96.80** | **94.04** | 38.55 | 52.69 | **24.76** | **27.09** | **28.65** |

An astronaut riding a horse

Darth vader surfing in waves

Figure 6: Minimizing the CD loss on VG data and maximizing reward objectives on WV data leads to color distortion in the generated videos.

## C.2 ABLATION STUDIES ON THE DATASETS FOR REWARD OPTIMIZATION

When training on the mixed dataset VG + WV, we propose minimizing CD loss using both VG and WV data and only maximizing the reward objectives Eq. 10 on the WV data with short captions. In this section, we further experiment with the other combination: 1) minimizing the CD loss on VG data and maximizing reward objectives on WV data. 2) minimizing the CD loss and maximizing reward objectives using both VG and WV data.

Our experiments show that the first setting led to reward over-optimization and color distortion in the generated videos, as shown in Fig. 6. Table 7 compares the second setting with the setting used in the main paper, demonstrating that optimizing rewards using both VG and WV reduces the benefit of aligning with RMs.

Table 7: Ablation studies on the datasets for reward optimization. While both settings use both VG and WV data to minimize CD loss, only leveraging WV data with short captions for reward optimization leads to better performance.

| Dataset for Optimizing Reward | Total Score | Quality Score | Subject Consist. | BG Consist. | Temporal Flickering | Motion Smooth. | Aesthetic Quality | Dynamic Degree | Image Quality |
|---|---|---|---|---|---|---|---|---|---|
| VG + WV | 82.38 | **84.35** | 96.74 | 97.15 | 97.93 | 96.91 | 65.52 | 69.17 | 71.14 |
| WV | **82.93** | 84.08 | 97.03 | 98.32 | 97.54 | 97.00 | 66.73 | 61.39 | 70.92 |

| Dataset for Optimizing Reward | Semantic Score | Object Class | Multiple Objects | Human Action | Color | Spatial Relation. | Scene | Appear. Style | Temporal Style | Overall Consist. |
|---|---|---|---|---|---|---|---|---|---|---|
| VG + WV | 74.49 | 94.59 | 49.21 | 95.00 | 93.98 | 42.59 | 51.31 | 23.87 | 26.09 | 28.18 |
| WV | **78.33** | 96.42 | 64.76 | 94.40 | 94.85 | 48.08 | 56.41 | 24.29 | 26.85 | 28.76 |

Table 8: Full results for the ablation studies on the design of training datasets, corresponding to Table 2 in the main paper. We bold the best results for each dimension and underline the second-best result.

| Models (Datasets) | Total Score | Quality Score | Subject Consist. | BG Consist. | Temporal Flicker. | Motion Smooth. | Aesthetic Quality | Dynamic Degree | Image Quality |
|---|---|---|---|---|---|---|---|---|---|
| **VCM** | | | | | | | | | |
| OpenVid | 78.52 | 83.62 | 96.87 | 96.95 | **98.33** | **97.92** | 63.59 | 53.06 | **71.98** |
| VidGen | 77.41 | 82.24 | 94.39 | 94.57 | 97.95 | 96.82 | 59.65 | 78.33 | 65.29 |
| WebVid | 76.15 | 81.31 | 92.82 | 94.18 | 96.58 | 96.65 | 57.24 | 84.72 | 65.05 |
| OV + WV | 73.30 | 80.00 | 93.82 | 95.03 | 96.07 | 97.20 | 54.48 | 61.94 | 67.88 |
| VG + WV | 78.49 | 82.95 | 94.83 | 95.84 | 97.42 | 97.17 | 60.92 | 75.56 | 67.98 |
| `T2V-Turbo-v2` w/o MG | | | | | | | | | |
| OpenVid | 80.97 | 84.04 | **97.50** | 98.02 | 98.28 | 97.82 | 64.43 | 55.83 | 70.78 |
| VidGen | 80.26 | 82.28 | 94.91 | 94.70 | 95.03 | 95.72 | 60.79 | **93.61** | 67.54 |
| WebVid | 81.34 | 83.41 | 97.32 | 98.29 | 98.13 | 97.16 | 65.36 | 48.89 | 71.75 |
| OV + WV | 81.00 | 82.32 | 97.27 | 97.22 | 97.60 | 97.47 | 62.76 | 45.28 | 70.93 |
| VG + WV | **82.93** | **84.08** | 97.03 | **98.32** | 97.54 | 97.00 | **66.73** | 61.39 | 70.92 |

| Models (Datasets) | Semantic Score | Object Class | Multiple Objects | Human Action | Color | Spatial Relation. | Scene | Appear. Style | Temporal Style | Overall Consist. |
|---|---|---|---|---|---|---|---|---|---|---|
| **VCM** | | | | | | | | | | |
| OpenVid | 61.93 | 79.92 | 20.12 | 90.40 | 88.14 | 26.59 | 28.21 | 23.44 | 23.22 | 26.24 |
| VidGen | 58.06 | 74.95 | 14.62 | 85.00 | 88.82 | 25.73 | 20.73 | 23.15 | 21.57 | 24.72 |
| WebVid | 55.51 | 62.52 | 10.15 | 82.40 | 86.22 | 20.98 | 18.39 | 23.78 | 23.11 | 24.86 |
| OV + WV | 46.53 | 34.35 | 3.99 | 64.20 | 90.60 | 19.67 | 5.94 | 22.97 | 21.34 | 21.72 |
| VG + WV | 60.65 | 76.46 | 16.14 | 85.20 | 85.45 | 28.76 | 28.76 | 23.35 | 23.90 | 26.02 |
| `T2V-Turbo-v2` | | | | | | | | | | |
| OpenVid | 68.73 | 91.20 | 33.96 | 93.60 | 92.99 | 30.83 | 41.70 | 23.59 | 24.77 | 27.16 |
| VidGen | 72.22 | 92.50 | 43.63 | 94.40 | 91.30 | 40.34 | 47.89 | 24.22 | 25.31 | 27.40 |
| WebVid | 73.04 | 91.74 | 47.41 | 93.60 | **96.53** | 42.02 | 44.16 | 24.06 | 25.65 | 28.28 |
| OV + WV | 75.74 | 92.50 | 55.00 | **95.40** | 95.02 | 36.47 | **57.53** | **24.73** | 26.56 | 28.31 |
| VG + WV | **78.33** | **96.42** | **64.76** | 94.40 | 94.85 | **48.08** | 56.41 | 24.29 | **26.85** | **28.76** |

Table 9: Full results for ablation studies on the RM design, corresponding to Table 9 in the main paper. We bold the best results for each dimension and underline the second-best result.

| Reward Models | Total Score | Quality Score | Subject Consist. | BG Consist. | Temporal Flicker. | Motion Smooth. | Aesthetic Quality | Dynamic Degree | Image Quality |
|---|---|---|---|---|---|---|---|---|---|
| VCM + $\mathcal{G}$ (No RM) | 79.02 | 82.78 | 95.28 | 95.38 | 97.00 | 97.35 | 60.12 | 76.39 | 67.84 |
| HPSv2.1 | 79.07 | 82.76 | 95.45 | 95.77 | 95.22 | 95.01 | 63.61 | 81.39 | 73.85 |
| + CLIP | 81.24 | 82.13 | 96.87 | 95.79 | 95.82 | 97.34 | 60.76 | 60.56 | 71.75 |
| + InternV2 | 82.02 | 84.17 | 97.18 | 99.07 | 97.33 | 97.03 | 67.67 | 58.61 | 71.26 |
| + CLIP + InternV2 | 83.52 | 85.13 | 95.50 | 96.71 | 97.35 | 97.07 | 62.61 | 90.00 | 71.78 |
| CLIP | 80.65 | 83.11 | 95.72 | 94.89 | 96.49 | 97.87 | 61.86 | 76.94 | 67.73 |
| + InternV2 | 82.04 | 84.05 | 96.60 | 97.70 | 97.94 | 98.15 | 62.91 | 66.67 | 68.24 |
| InternV2 | 81.37 | 83.02 | 96.35 | 97.01 | 96.66 | 97.30 | 65.74 | 57.50 | 70.89 |

| Reward Models | Semantic Score | Object Class | Multiple Objects | Human Action | Color | Spatial Relation. | Scene | Appear. Style | Temporal Style | Overall Consist. |
|---|---|---|---|---|---|---|---|---|---|---|
| VCM + $\mathcal{G}$ (No RM) | 64.01 | 83.61 | 23.12 | 89.20 | 75.92 | 33.46 | 38.90 | 23.55 | 24.95 | 26.39 |
| HPSv2.1 | 64.28 | 84.95 | 20.84 | 88.40 | 83.41 | 27.43 | 43.20 | 22.98 | 24.59 | 26.56 |
| + CLIP | 77.66 | 96.90 | 63.26 | 93.80 | 94.36 | 50.97 | 52.91 | 24.37 | 26.55 | 28.05 |
| + InternV2 | 73.40 | 94.19 | 47.70 | 93.40 | 90.69 | 40.19 | 53.30 | 23.59 | 25.70 | 27.80 |
| + CLIP + InternV2 | 77.12 | 95.33 | 61.49 | 96.20 | 92.53 | 43.32 | 56.40 | 24.17 | 27.06 | 28.26 |
| CLIP | 70.80 | 91.53 | 40.08 | 93.00 | 91.98 | 36.11 | 42.21 | 24.62 | 25.89 | 27.61 |
| + InternV2 | 74.01 | 94.51 | 47.47 | 96.40 | 94.25 | 36.47 | 50.71 | 24.23 | 26.43 | 28.36 |
| InternV2 | 74.75 | 95.57 | 43.55 | 96.80 | 94.95 | 41.64 | 56.41 | 24.12 | 25.86 | 27.72 |

Table 10: Full results for ablation studies on the effectiveness of motion guidance, corresponding to Table 4 in the main paper.

| Models (Datasets) | Total Score | Quality Score | Subject Consist. | BG Consist. | Temporal Flicker. | Motion Smooth. | Aesthetic Quality | Dynamic Degree | Image Quality |
|---|---|---|---|---|---|---|---|---|---|
| T2V-Turbo (OV+WV) | 81.00 | 82.32 | 97.27 | 97.22 | 97.60 | 97.47 | 62.76 | 45.28 | 70.93 |
| T2V-Turbo (VG+WV) | 82.93 | 84.08 | 97.03 | 98.32 | 97.54 | 97.00 | 66.73 | 61.39 | 70.92 |
| T2V-Turbo-v2 (OV+WV) | 81.81 | 83.15 | 97.18 | 97.81 | 97.10 | 96.05 | 66.08 | 60.28 | 71.04 |
| T2V-Turbo-v2 (VG+WV) | 83.52 | 85.13 | 95.50 | 96.71 | 97.35 | 97.07 | 62.61 | 90.00 | 71.78 |

| Models (Datasets) | Semantic Score | Object Class | Multiple Objects | Human Action | Color | Spatial Relation. | Scene | Appear. Style | Temporal Style | Overall Consist. |
|---|---|---|---|---|---|---|---|---|---|---|
| T2V-Turbo (OV+WV) | 75.74 | 92.50 | 55.00 | 95.40 | 95.02 | 36.47 | 57.53 | 24.73 | 26.56 | 28.31 |
| T2V-Turbo (VG+WV) | 78.33 | 96.42 | 64.76 | 94.40 | 94.85 | 48.08 | 56.41 | 24.29 | 26.85 | 28.76 |
| T2V-Turbo-v2 (OV+WV) | 76.47 | 94.48 | 53.54 | 96.60 | 94.10 | 42.35 | 56.89 | 24.48 | 26.47 | 28.94 |
| T2V-Turbo-v2 (VG+WV) | 77.12 | 95.33 | 61.49 | 96.20 | 92.53 | 43.32 | 56.40 | 24.17 | 27.06 | 28.26 |

Table 11: **Automatic evaluation results on VBench** (Huang et al., 2024). We compare our `T2V-Turbo-v2` with baseline methods across the 16 VBench dimensions. A higher score indicates better performance for a particular dimension. We bold the best results for each dimension and underline the second-best result. **Quality Score** is calculated with the 7 dimensions f rom the top table. **Semantic Score** is calculated with the 9 dimensions from the bottom table. **Total Score** a weighted sum of **Quality Score** and **Semantic Score**. Our `T2V-Turbo-v2` **surpass all baseline methods with 8 inference steps** in terms of Total Score, Quality Score, and Semantic Score, including the proprietary systems Gen-3 and Kling.

| Models | Total Score | Quality Score | Subject Consist. | BG Consist. | Temporal Flicker. | Motion Smooth. | Aesthetic Quality | Dynamic Degree | Image Quality |
|---|---|---|---|---|---|---|---|---|---|
| VideoCrafter2 | 80.44 | 82.20 | 96.85 | 98.22 | 98.41 | 97.73 | 63.13 | 42.50 | 67.22 |
| Pika | 80.40 | 82.68 | 96.76 | **98.95** | **99.77** | 99.51 | 63.15 | 37.22 | 62.33 |
| Gen-2 | 80.58 | 82.47 | 97.61 | 97.61 | 99.56 | **99.58** | **66.96** | 18.89 | 67.42 |
| Gen-3 | 82.32 | 84.11 | 97.10 | 96.62 | 98.61 | 99.23 | 63.34 | 60.14 | 66.82 |
| Kling | 81.85 | 83.39 | **98.33** | 97.60 | 99.30 | 99.40 | 46.94 | 61.21 | 65.62 |
| T2V-Turbo | 81.01 | 82.57 | 96.28 | 97.02 | 97.48 | 97.34 | 63.04 | 49.17 | **72.49** |
| `T2V-Turbo-v2` | | | | | | | | | |
| w/o MG | **82.93** | 84.08 | 97.03 | **98.32** | 97.54 | 97.00 | **66.73** | 61.39 | 70.92 |
| w/ MG | **83.52** | **85.13** | 95.50 | 96.71 | 97.35 | 97.07 | 62.61 | 90.00 | 71.78 |

| Models | Semantic Score | Object Class | Multiple Objects | Human Action | Color | Spatial Relation. | Scene | Appear. Style | Temporal Style | Overall Consist. |
|---|---|---|---|---|---|---|---|---|---|---|
| VideoCrafter2 | 73.42 | 92.55 | 40.66 | 95.00 | 92.92 | 35.86 | 55.29 | **25.13** | 25.84 | 28.23 |
| Pika | 71.26 | 87.45 | 46.69 | 88.00 | 85.31 | 65.65 | 44.80 | 21.89 | 24.44 | 25.47 |
| Gen-2 | 73.03 | 90.92 | 55.47 | 89.20 | 89.49 | 66.91 | 48.91 | 19.34 | 24.12 | 26.17 |
| Gen-3 | 75.17 | 87.81 | 53.64 | **96.40** | 80.90 | 65.09 | 54.57 | 24.31 | 24.71 | 26.69 |
| Kling | 75.68 | 87.24 | **68.05** | 93.40 | 89.90 | **73.03** | 50.86 | 19.62 | 24.17 | 26.42 |
| T2V-Turbo | 74.76 | **93.96** | 54.65 | 95.20 | 89.90 | 38.67 | **55.58** | 24.42 | 25.51 | 28.16 |
| `T2V-Turbo-v2` | | | | | | | | | | |
| w/o MG | **78.33** | **96.42** | **64.76** | 94.40 | 94.85 | **48.08** | 56.41 | 24.29 | 26.85 | 28.76 |
| w/ MG | 77.12 | 95.33 | 61.49 | 96.2 | 92.53 | 43.32 | 56.4 | 24.17 | 27.06 | 28.26 |

## D ADDITIONAL QUALITATIVE RESULTS

**Videos: click to play**

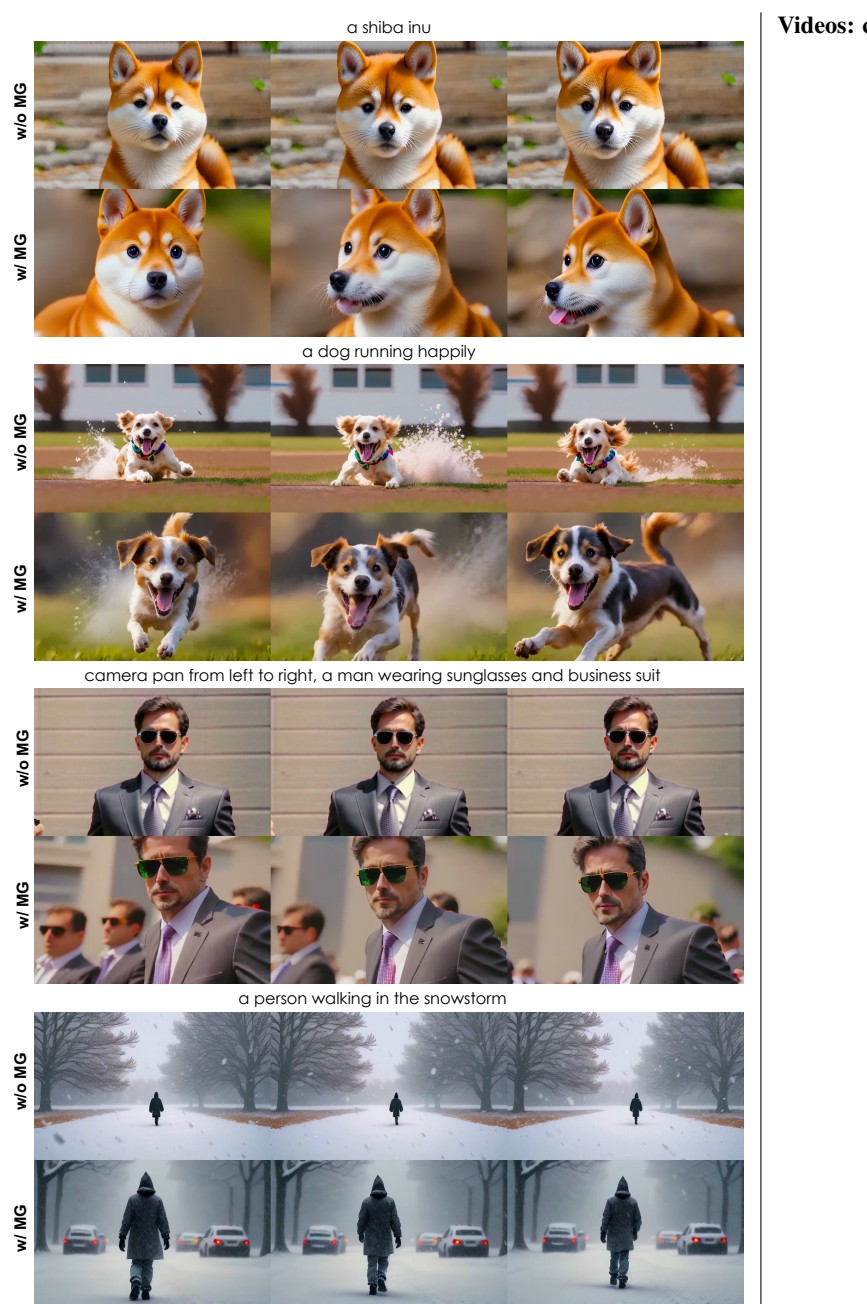

Figure 7: Additional qualitative comparison between `T2V-Turbo-v2` w/o MG and w/ MG. Integrating the Motion guidance leads to richer video motion that aligns with the prompt. **Play videos in Adobe Acrobat**.

**Videos: click to play**

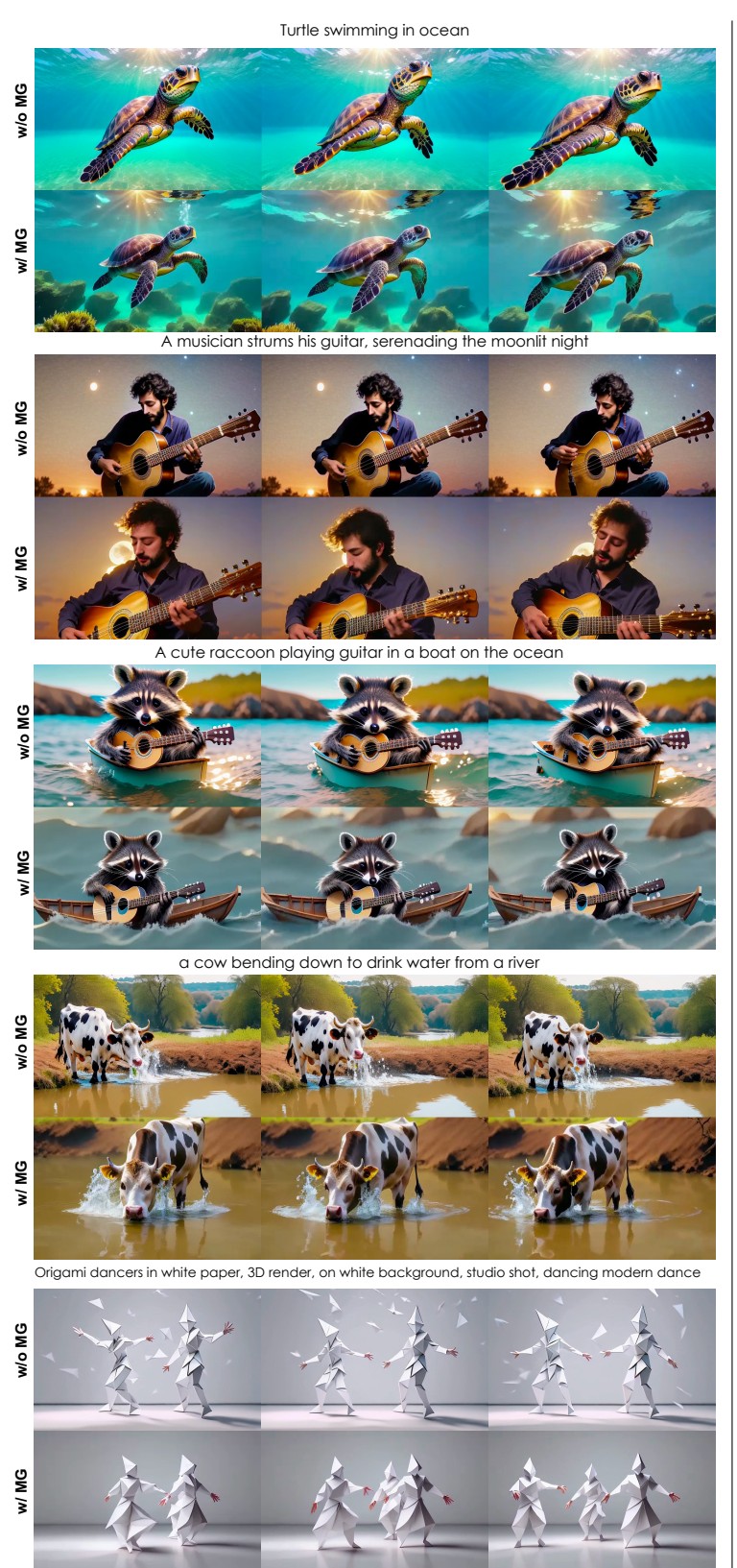

Figure 8: Additional qualitative comparison between `T2V-Turbo-v2` w/o MG and w/ MG. Integrating the Motion guidance leads to richer video motion that aligns with the prompt.

# E ADDITIONAL ABLATION STUDIES ON THE DESIGN OF TRAINING DATASETS

Table 12: Ablation studies on the design of training datasets. Based on the results in Table 2, we further conduct experiments on OV + VG and OV + VG + WV to corroborate the results.

| | VCM | | | | | | | T2V-Turbo-v2 w/o MG | | | | | | |
|---|---|---|---|---|---|---|---|---|---|---|---|---|---|---|
| | OV | VG | WV | OV + WV | VG + WV | OV + VG | OV + VG + WV | OV | VG | WV | OV + WV | VG + WV | OV + VG | OV + VG + WV |
| Quality Score | **83.62** | 82.24 | 81.31 | 80.00 | 82.95 | 83.43 | 82.95 | 84.04 | 82.28 | 83.41 | 82.32 | **84.08** | 83.86 | 82.35 |
| Semantic Score | **61.93** | 58.06 | 55.51 | 46.53 | 60.65 | 57.52 | 54.98 | 68.73 | 72.22 | 73.04 | 75.74 | **78.33** | 69.25 | 76.80 |
| Total Score | **78.52** | 77.41 | 76.15 | 73.30 | 78.49 | 78.25 | 77.36 | 80.97 | 80.26 | 81.34 | 81.00 | **82.93** | 80.94 | 81.24 |

We further evaluate the performance of conduct experiments on the OV + VG and OV + VG + WV datasets to corroborate our conclusions in Table 2 in Sec. 4.2.

As shown in the table, VCM achieves a high Quality Score on the OV + VG dataset, similar to training on pure OV data, but adding the lower-quality WV data slightly decreases this score. Conversely, incorporating reward feedback in T2V-Turbo-v2 significantly improves the Semantic Score for the OV + VG + WV dataset, while the gains for OV + VG remain comparatively moderate. These findings align with our discussion in the main text: RMs with short context lengths operate optimally on datasets with shorter captions, highlighting the importance of aligning dataset characteristics with RM capabilities. Furthermore, the results justify our decision to exclude OV data when training the main T2V-Turbo-v2 models, as incorporating OV into VG + WV datasets negatively impacts model performance.

Figure 9: Human evaluation on 16-step generation of T2V-Turbo-v2 w/o MG and T2V-Turbo-v2 w/ MG shows a clear preference for the latter. By incorporating motion information extracted from the training videos, T2V-Turbo-v2 w/ MG consistently produces videos that are favored for their superior motion quality and overall appeal.

# F  COMPARING OUR `T2V-TURBO-V2` WITH T2V-TURBO VIA HUMAN EVALUATION

We conduct a human evaluation to compare the 16-step video generation of T2V-Turbo-v2 w/o MG and T2V-Turbo-v2 w/ MG to verify the effectiveness of motion guidance. We carefully select 50 prompts that explicitly require motion generation from VBench. Each method generates 5 videos for each prompt with the same set of random seeds.

We hire annotators from Amazon Mechanical Turk to answer two questions: Q1) Which video demonstrates better motion quality? Q2) Which video do you prefer given the prompt?

We form the video comparison task as many batches of HITs. To ensure the annotation quality, we ensure the annotators are from English-speaking countries, including AU, CA, NZ, GB, and the US. Each task needs around 30 seconds to complete, and we pay each submitted HIT with 0.2 US dollars. Therefore, the hourly payment is about 24 US dollars. Note that the data annotation part of our project is classified as exempt by Human Subject Committee via IRB protocols.

The human evaluation results in Figure 9 show that videos generated by T2V-Turbo-v2 w/ MG are consistently preferred over those from T2V-Turbo w/o MG in terms of motion quality and overall appeal. These findings corroborate our automatic evaluation in Table 4, confirming that incorporating motion guidance significantly enhances model performance and improves the motion quality of generated videos.

