# OpenReview forum: "T2V-Turbo-v2: Enhancing Video Model Post-Training through Data, Reward, and Conditional Guidance Design"
_ICLR.cc/2025/Conference — ICLR 2025 Poster_

### Official Review · Reviewer_XVsE · 2024-10-28

**Soundness:** 4
**Presentation:** 3
**Contribution:** 3
**Rating:** 6
**Confidence:** 4

**Summary:**

In the paper, the authors present T2V-Turbo-v2, a method to enhance diffusion-based text-to-video models by distilling a consistency model from a pre-trained T2V model. This approach integrates high-quality training data, feedback from multiple reward models, and motion guidance into the distillation process. Through ablation studies, it emphasizes the importance of high-quality datasets and diverse reward models to improve visual quality and text-video alignment. The method also verifies the effectiveness of incorporating motion guidance to enhance video motion quality. T2V-Turbo-v2 achieves a state-of-the-art total score of 85.13 on VBench, outperforming advanced text-to-video models like Gen-3 and Kling.

**Strengths:**

1. The experiments are comprehensive and thorough, with detailed analysis.
2. The analysis of minimizing CD loss using entire datasets while restricting reward optimization to short-captioned datasets is interesting and meaningful, which may encourage future work.
3. This paper establishes a new SOTA total score on VBench, leveraging open-source models to outperform some advanced text-to-video models.
4. The paper is well-written and easy to understand.

**Weaknesses:**

1. The integration of MotionClone into T2V-Turbo is an interesting direction. However, while the contribution highlights the potential for diverse forms of energy functions, this study primarily utilizes the motion representation from the MotionClone work without substantial modifications to the energy function's format. The other enhancements are relatively minor, such as the reward model used, which only adds a CLIP compared to T2V-Turbo, and the removal of the EMA model. It might be beneficial to explore further variations to strengthen the contribution.
2. It would be beneficial to include a discussion on the performance differences between VCM and the proposed method across various datasets. For instance:
    - Could you explore why VCM achieves the best performance using only OV, while the proposed method does not attain similar results? It appears that different methods may lead to varying conclusions regarding dataset choices. Additionally, how would the results differ if T2V-Turbo (not v2) was used?
    - Considering OV and VG are both high-quality video datasets, it would be insightful to analyze why OV+WV exhibits poorer performance compared to VG+WV, which performs quite well.
3. In the section on data preprocessing, the approach involves using DDIM Inversion on all videos to obtain the necessary motion guidance for training, which is effective in reducing training time. Nevertheless, this approach does not significantly simplify the overall complexity. It would be better to explore improvements to the motion guidance strategy itself to enhance training efficiency.
4. It would be valuable to include theoretical or experimental results to analyze why the EMA model in consistency distillation is unnecessary.
5. It would be better to conduct a user study to further verify the performance from a human perspective.

**Questions:**

1. The pseudo-code in Algorithm 2 for training includes theta-, but the method states that EMA is not needed. Please update either the text or the algorithm to ensure consistency.

---

> ### Author Response · Authors · 2024-11-21
> **(1/3) Response to Reviewer XVsE**
>
> We thank the reviewer for the constructive feedback. Please find our detailed response below.
>
> > The integration of MotionClone into T2V-Turbo is an interesting direction. However, while the contribution highlights the potential for diverse forms of energy functions, this study primarily utilizes the motion representation from the MotionClone work without substantial modifications to the energy function's format.
>
> We would like to clarify some misunderstandings in the reviewer’s assessment of our work:
>
> 1. **Objective of the Study**: The primary goal of our work is not to design a new energy function for motion guidance. Instead, we aim to empirically demonstrate that augmenting the teacher ODE solver with the energy function's gradients of a conditional guidance strategy can **distill a more capable student video generator while significantly reducing inference costs**. In this context, we showcase the potential of our approach by integrating MotionClone’s energy function into the teacher ODE solver. In other words, **our method is not confined to MotionClone’s motion guidance**.
> 2. **Challenges in Scaling MotionClone for Training**: Adapting MotionClone’s motion guidance techniques for training is significantly non-trivial for several reasons:
>     - **Dependency on Reference Videos**: MotionClone relies on access to reference videos with high-quality motion. However, identifying suitable reference videos for general text prompts is challenging, which limits its effectiveness and applicability for generic video generation tasks.
>     - **High Computational Cost**: Computing the gradient of the energy function during inference incurs substantial memory and latency overhead. For instance, generating a single video with MotionClone can take approximately seven minutes and 30 GB GPU memory.
>
>     To address these challenges, we leverage the critical insight that each training video inherently serves as an ideal reference video for its corresponding training prompt. Additionally, we design a separate preprocessing phase to precompute the motion guidance before the actual training phase. As a result, this preprocessing phase eliminates the need for computationally expensive gradient calculations during training.
>
> > The other enhancements are relatively minor, such as the reward model used, which only adds a CLIP compared to T2V-Turbo.
> >
>
> We would like to clarify the contributions of our work and address the concerns regarding the significance of our enhancements.
>
> 1. **Coupled Effects of Training Data and Reward Models**:
>
>     The choices of training data and reward models are arguably the most critical components in the post-training phase of a generative AI model. In our work, we conduct a rigorous and thorough empirical investigation into how these factors impact the performance of T2V models. A key finding of our study is that their effects are **not orthogonal**—the interaction between training datasets and RMs plays a pivotal role in shaping the final performance. Specifically, our Section 4.2, 4.3, and Appendix C.2 empirically demonstrate that **curating training datasets for different learning objectives is crucial** for achieving optimal results. To the best of our knowledge, this is the first work to systematically study how the selection of training data and RMs affects the post-training performance of a T2V model. Therefore, we firmly believe our findings provide invaluable insights for advancing post-training research in video generation models.
>
> 2. **Different Conclusion Regarding the Use of Reward Models**:
>
>     Our findings regarding the effects of different RMs diverge significantly from the conclusions of the T2V-Turbo paper. T2V-Turbo suggests that feedback from a single image-text RM (HPSv2.1) is sufficient to achieve substantial performance gains. In contrast, our work reveals that relying solely on HPSv2.1 results in only minimal enhancements to video quality. Instead, we show that incorporating feedback from a more diverse set of RMs is essential to achieve meaningful performance improvements.
>
>     The reasons behind these differing conclusions are centered on the datasets used for different learning objectives, which are summarized below:
>
>     - Our T2V-Turbo-v2 leverages a dataset that combines videos from VG and WV. Specifically, we minimize the consistency distillation loss on the entire dataset but optimize reward objectives only on the short-captioned WV data.
>     - T2V-Turbo was trained exclusively on WV data with short video captions.
>
>     These experimental differences underscore a critical insight: the impact of reward feedback is highly dependent on the dataset composition and design. Our results highlight the importance of curating datasets and carefully selecting RM sets to achieve optimal performance in video generation tasks.

---

> > ### Author Response · Authors · 2024-11-21
> > **(2/3) Response to Reviewer XVsE**
> >
> > > Could you explore why VCM achieves the best performance using only OV, while the proposed method does not attain similar results?
> >
> > The performance difference is discussed in Lines 365–375. We hypothesize that the modest performance gains of T2V-Turbo-v2 on OV stem from the excessively long captions in the OV dataset, which are not well-suited to the short context length of the RMs we used. Specifically, the maximum context length of HPSv2.1 and CLIP is 77 tokens, while InternV2’s is only 40 tokens. Consequently, these RMs can only operate optimally on datasets with shorter captions, which explains why T2V-Turbo-v2 performs better on WV compared to OV or VG.
> >
> > In contrast, VCM optimizes only the consistency distillation loss, making it better equipped to leverage the high-quality video data in OV, resulting in its superior performance on this dataset.
> >
> > > It appears that different methods may lead to varying conclusions regarding dataset choices.
> > >
> >
> > We highlight this point through out our paper. Curating training datasets for different learning objective is crucial to achieve optimal results.
> >
> > > Additionally, how would the results differ if T2V-Turbo (not v2) was used?
> >
> > We believe that similar results will be obtained if we use T2V-Turbo that optimizes the LoRA weights, as the results in Table 2 are obtained without augmenting the teacher ODE solver with motion guidance (w/o MG).
> >
> > > Considering OV and VG are both high-quality video datasets, it would be insightful to analyze why OV+WV exhibits poorer performance compared to VG+WV, which performs quite well.
> >
> > We conjecture that the poorer performance of VCM and our T2V-Turbo-v2 on the OV+WV dataset stems from the significant domain gap between OV and WV data. In our preliminary study, we observed that most OV videos are centered around human activities, whereas both WV and VG datasets encompass a broader diversity of video content. Although WV videos are lower in quality and often include watermarks, their content diversity aligns more closely with VG than OV. This domain gap likely hampers the performance of VCM and T2V-Turbo-v2 on OV+WV, whereas the VG+WV combination benefits from greater alignment between the datasets.
> >
> > > In the section on data preprocessing, the approach involves using DDIM Inversion on all videos to obtain the necessary motion guidance for training, which is effective in reducing training time. Nevertheless, this approach does not significantly simplify the overall complexity. It would be better to explore improvements to the motion guidance strategy itself to enhance training efficiency.
> >
> > We acknowledge that our preprocessing step requires additional computational resources, amounting to approximately 400 GPU hours. However, this is a manageable cost that can be completed in about two days on a server equipped with 8 NVIDIA A100 GPUs (40 GB each). Importantly, this additional preprocessing effort yields significant benefits in terms of motion quality and inference acceleration.
> >
> > In comparison, the original MotionClone approach requires approximately 6 minutes to generate a single video, with 3 minutes spent on DDIM inversion from a reference video. Moreover, the peak GPU memory consumption can reach up to 35 GB due to the gradient calculations involved. In contrast, our model eliminates the need for identifying appropriate reference videos, conducting DDIM inversion, or calculating gradients during inference without adding any inference overhead compared to inference from a VCM. For example, our T2V-Turbo-v2 only takes 5 seconds to generate a video in 16 steps using BFloat16.
> >
> > The efficiency and performance gains of our approach are clearly reflected in its results. Our model achieves the #1 ranking on VBench, surpassing numerous proprietary video generation systems, including Gen-3, Kling, and MiniMax. This demonstrates that the preprocessing trade-off is well-justified, offering both superior motion quality and significant reductions in inference time.
> >
> > > It would be valuable to include theoretical or experimental results to analyze why the EMA model in consistency distillation is unnecessary.
> >
> > Our decision to remove the EMA model is based on empirical observations, which show that its removal does not lead to training instability. For readers seeking a theoretical perspective, we refer to Section 3.2 of [1], which provides an in-depth analysis explaining why the EMA model can be omitted without compromising training stability.
> >
> > [1] Song et al. Improved techniques for training consistency models. ICLR 2024

---

> > > ### Author Response · Authors · 2024-11-21
> > > **(3/3) Response to Reviewer XVsE**
> > >
> > > > It would be better to conduct a user study to further verify the performance from a human perspective.
> > >
> > > We thank the reviewer for the constructive suggestion. In response, we conducted a human evaluation to compare the 16-step video generation of T2V-Turbo-v2 w/o MG and T2V-Turbo-v2 w/ MG to verify the effectiveness of motion guidance.
> > >
> > > We hire annotators from Amazon Mechanical Turk to answer two questions: Q1) Which video demonstrates better motion quality? Q2) Which video do you prefer given the prompt? Appendix E provides further experimental details.
> > >
> > > The human evaluation results in Figure 9 of Appendix F show that videos generated by T2V-Turbo-v2 w/ MG are consistently favored over those from T2V-Turbo in terms of motion quality and general preference. These findings corroborate our automatic evaluation in Table 4, verifying that incorporating motion guidance enhances model performance and improves the motion quality of the generated videos.
> > >
> > > > The pseudo-code in Algorithm 2 for training includes theta-, but the method states that EMA is not needed. Please update either the text or the algorithm to ensure consistency.
> > >
> > > We thank the reviewer for pointing it out. We have updated Algorithm 2 to remove $\theta^-$.

---

> ### Author Response · Authors · 2024-11-24
> **Looking forward to your response!**
>
> Dear Reviewer XVsE,
>
> We greatly appreciate your insightful feedback, which has significantly contributed to the clarity and enhancement of our work. We have carefully addressed your comments in our response, clarified potential misunderstandings, and included the new human evaluation results you requested.
>
> We kindly invite you to revisit our paper in light of these updates and clarifications. We would greatly appreciate it if you could consider whether these changes warrant a reevaluation of your rating.
>
> Best regards,
>
> The Authors

---

> > ### Author Response · Authors · 2024-11-26
> > **Follow up the discussion**
> >
> > Dear Reviewer XVsE,
> >
> > Thank you again for serving as a reviewer! Your feedback has been valuable in helping us clarify, improve, and refine our work. We have worked diligently to address your comments and included new human evaluation results.
> >
> > We would like to invite you to continue the discussion with us. We hope that our responses can successfully address your concerns so that you might consider a reevaluation of your rating.
> >
> > Thanks and best regards,
> >
> > The Authors

---

> > > ### Author Response · Authors · 2024-11-27
> > > **Let's discuss!**
> > >
> > > Dear Reviewer XVsE,
> > >
> > > Thank you once again for your time and efforts in providing valuable feedback on our work. Your insights have been instrumental in helping us refine and improve our submission.
> > >
> > > We would like to kindly invite you to follow up on the discussion regarding our work. If you have any additional comments or concerns, please don’t hesitate to let us know, and we will do our utmost to address them promptly.
> > >
> > > The Authors

---

> > > > ### Author Response · Authors · 2024-11-28
> > > > **Happy Thanksgiving!**
> > > >
> > > > Dear Reviewer XVsE,
> > > >
> > > > On this Thanksgiving, we would like to take the opportunity to express our heartfelt gratitude for your time and effort in providing valuable feedback on our work. Your insights have been truly invaluable in helping us refine and improve our submission.
> > > >
> > > > We kindly invite you to follow up on the discussion regarding our work. Should you have any additional comments or concerns, please don’t hesitate to let us know—we are committed to addressing them to the best of our ability.
> > > >
> > > > The Authors

---

> > > > > ### Comment · Reviewer_XVsE · 2024-11-29
> > > > >
> > > > > I sincerely thank the authors for their responses and the additional experiments they provided. I reviewed the comments from other reviewers, and the authors have addressed most of my concerns.
> > > > >
> > > > > However, despite the further clarification on contributions, I still believe that the contributions are relatively limited.
> > > > >
> > > > > 1. As reviewers PqeK and dKgW pointed out, the methodology appears to be a combination of MotionClone and consistency distillation. Although the authors mentioned that they aim to validate the enhancement of video generation models using the energy function's gradients of a conditional guidance strategy, relying solely on MotionClone as a contribution seems insufficient, especially since the paper uses MotionClone's guidance strategy without modifications or enhancements.
> > > > >
> > > > > 2. The authors discussed challenges in scaling MotionClone for training.
> > > > >    - Regarding the dependency on reference videos, MotionClone itself does not use carefully curated high-quality videos but rather employs commonly used videos like those from the DAVIS dataset. Hence, the idea of using training videos as their own reference videos is natural and not particularly captivating.
> > > > >    - Regarding the computational cost, the authors mention that the proposed method does not increase inference cost. Does this imply that motion guidance is only required during training and not during inference?
> > > > >
> > > > > 3. While the selection and analysis of training data and reward models are interesting, the authors conducted experiments and analyses only on VideoCrafter2 without attempting further validation on more models. Thus, it remains unclear whether the conclusions are applicable to other models.

---

> > > > > > ### Author Response · Authors · 2024-11-29
> > > > > > **Response to Reviewer XVsE**
> > > > > >
> > > > > > Dear Reviewer XVsE,
> > > > > >
> > > > > > We appreciate your detailed feedback. Please find our detailed comments below.
> > > > > >
> > > > > > > The methodology appears to be a combination of MotionClone and consistency distillation. Relying solely on MotionClone as a contribution seems insufficient, especially since the paper uses MotionClone's guidance strategy without modifications or enhancements.
> > > > > >
> > > > > > We would like to address a major misunderstanding by the reviewer. **We do not rely on MotionClone as a contribution.** Instead, our work demonstrates how conditional guidance strategies—specifically, the gradients of energy functions—can be utilized as additional supervision signals during the post-training phase of video generation models.
> > > > > >
> > > > > > In this paper, we adapt MotionClone's guidance strategy to the training phase by addressing two key challenges: (1) its reliance on manually selected reference videos and (2) its substantial computational cost. Moreover, **our method does not introduce inference overhead, while MotionClone requires approximately 7 minutes to generate a video.**
> > > > > >
> > > > > > Again, we emphasize that our method is not limited to MotionClone’s motion guidance but highlights the broader potential of conditional guidance strategies to enhance model performance.
> > > > > >
> > > > > > > MotionClone itself does not use carefully curated high-quality videos but rather employs commonly used videos like those from the DAVIS dataset. Hence, the idea of using training videos as their own reference videos is natural and not particularly captivating.
> > > > > >
> > > > > > The original DAVIS [1] dataset contains only 50 videos, with MotionClone utilizing just 40 of them. This severely limits the scalability of MotionClone for training, as it is impractical to rely on such a small dataset. Additionally, **MotionClone requires manual effort to select different reference videos for different text prompts**, further complicating its application at scale.
> > > > > >
> > > > > > In contrast, our approach is designed to handle datasets with millions of videos and eliminates the need for any manual effort in selecting reference videos for text prompts. By leveraging each training video as its own reference, our method achieves scalability and automation, addressing the challenges inherent in MotionClone’s original setup.
> > > > > >
> > > > > > [1] Pont-Tuset et al., The 2017 DAVIS Challenge on Video Object Segmentation.
> > > > > >
> > > > > > > Does this imply that motion guidance is only required during training and not during inference?
> > > > > >
> > > > > > We never use MotionClone's guidance during inference. Our generation process is as fast as the baseline VCM.
> > > > > >
> > > > > > > Thus, it remains unclear whether the conclusions are applicable to other models.
> > > > > >
> > > > > > The original MotionClone is built on top of AnimateDiff [2], and prior work, such as AnimateLCM [3], has demonstrated that an LCM can be learned from AnimateDiff. Consequently, there is no inherent barrier to applying our approach to other models, such as AnimateDiff.
> > > > > >
> > > > > > We argue that **our experiments already highlight the strong generalizability of our method**. Unlike the original MotionClone paper, which uses AnimateDiff as its base model, we chose VideoCrafter2 as our foundation, successfully demonstrating the applicability of our approach to different video base models.
> > > > > >
> > > > > > [2] Guo et al., AnimateDiff: Animate Your Personalized Text-to-Image Diffusion Models without Specific Tuning. ICLR 2024.
> > > > > >
> > > > > > [3] Wang et al., AnimateLCM: Computation-Efficient Personalized Style Video Generation without Personalized Video Data. SIGGRAPH ASIA 2024

---

> > > > > > > ### Comment · Reviewer_XVsE · 2024-11-30
> > > > > > >
> > > > > > > Thank the authors for their timely response, which has partially addressed my concerns.
> > > > > > >
> > > > > > > The authors have highlighted that the paper aims to validate the enhancement of video generation models using the energy function's gradients of a conditional guidance strategy, and that the method is not limited to MotionClone’s motion guidance. However, I believe that simply applying MotionClone and integrating it in a straightforward manner is relatively insufficient to enhance the novelty of the method and validate the research objective. Thus, I remain unconvinced regarding the methodological contributions.
> > > > > > >
> > > > > > > I realize I may not have expressed this clearly before, so I would like to further clarify my point about reliance on MotionClone as a contribution: since the authors mentioned that the approach is not limited to MotionClone's motion guidance, it would be better either to experiment with multiple conditional guidance strategies or to further improve MotionClone. This would better highlight the innovation of the method and the persuasiveness of the research objective.
> > > > > > >
> > > > > > > Additionally, the authors noted that the proposed method does not require motion guidance during inference. I believe this is valuable and meaningful for the field of video generation.
> > > > > > >
> > > > > > > Overall, after reconsidering the strengths and weaknesses of this paper in light of my concerns, I have decided to raise my score to 6 and leave the final decision to the area chair.

---

> > > > > > > > ### Author Response · Authors · 2024-11-30
> > > > > > > > **Response to Reviewer XVsE**
> > > > > > > >
> > > > > > > > Dear Reviewer XVsE,
> > > > > > > >
> > > > > > > > Thank you so much for raising your score! Please find our detailed response below.
> > > > > > > >
> > > > > > > > > I believe that simply applying MotionClone and integrating it in a straightforward manner is relatively insufficient to enhance the novelty of the method and validate the research objective.
> > > > > > > >
> > > > > > > > First, we would like to reiterate that the primary goal of our research is not to design a superior energy function to improve motion guidance. Instead, we aim to highlight the vast design space of conditional strategies and demonstrate their potential to enable a more capable student video generator without adding inference overhead. In this paper, we leverage MotionClone's energy function to augment the teacher ODE solver, demonstrating its effectiveness in enhancing the motion quality of the generated videos.
> > > > > > > >
> > > > > > > > Second, integrating MotionClone into the training process is far from straightforward. As highlighted in the paper and our earlier response, scaling MotionClone for training involves significant engineering efforts to address its considerable computational cost. **We believe our paper successfully achieves its proof-of-concept mission, paving the way for future research to distill knowledge from a more diverse set of energy functions.**
> > > > > > > >
> > > > > > > >
> > > > > > > > > it would be better either to experiment with multiple conditional guidance strategies.
> > > > > > > >
> > > > > > > > We sincerely thank the reviewer for their valuable suggestion. We agree that experimenting with multiple conditional guidance strategies would further strengthen our argument. However, due to the limited timeframe of the rebuttal period, we are unable to conduct these additional experiments and plan to explore them in future work.

---

### Official Review · Reviewer_dKgW · 2024-10-28

**Soundness:** 3
**Presentation:** 3
**Contribution:** 2
**Rating:** 6
**Confidence:** 5

**Summary:**

This paper present motionclone-based consistency distillation, using motion guidance to improve temporal and spatial coherence.

**Strengths:**

1. This method demonstrates strong performance, achieving state-of-the-art (SOTA) results on VBench. Visualized video outputs appear smooth and high-quality, reflecting its effective design.

2. The paper is clearly written, allowing reviewers to easily understand the authors' intent.

3. This method is simple and effective, making it generally more practical.

**Weaknesses:**

This method is overly engineering-focused and lacks novelty, as the motion guidance and consistency distillation techniques involved are already established, making it appear less innovative.

Additionally, while it conducts extensive ablation experiments on motion guidance and reward models, this does not constitute a significant contribution of the paper. I am unclear about the paper's contributions; is it providing more interesting insights? It would be helpful if the authors could briefly summarize this in their response.

Regarding the contribution summary of the paper (L117), it seems to emphasize the advantages of existing work and the potential of extracting motion priors. And, I do not see any strong insights that stand out; motionclone has already demonstrated this fairly clearly. If the authors mean that motion priors are particularly useful during T2V training, they should provide more experiments. For example, training SVD and VideoCrafter2 shows that the insights presented in T2V-Turbo are quite limited.

**Questions:**

1. I’m curious why this method, although based on distilling VideoCrafter2, outperforms VideoCrafter2 across multiple tasks, such as in the three metrics shown in Table 1.

---

> ### Author Response · Authors · 2024-11-21
> **(1/2) Response to Reviewer dKgW**
>
> > This method is overly engineering-focused and lacks novelty, as the motion guidance and consistency distillation techniques involved are already established, making it appear less innovative.
>
> We respectfully disagree with the reviewer's argument on our novelty. In terms of guidance strategy, the primary goal of our work is not to design a new energy function for motion guidance. Instead, we aim to empirically demonstrate that augmenting the teacher ODE solver with the energy function's gradients of a conditional guidance strategy can distill a more capable student video generator while significantly reducing inference costs. In this context, we showcase the potential of our approach by integrating MotionClone’s energy function into the teacher ODE solver.
>
> Additionally, adapting MotionClone’s motion guidance techniques for training is significantly non-trivial for several reasons:
>
> - **Dependency on Reference Videos**: MotionClone relies on access to reference videos with high-quality motion. However, identifying suitable reference videos for general text prompts is challenging, which limits its effectiveness and applicability for generic video generation tasks.
> - **High Computational Cost**: Computing the gradient of the energy function during inference incurs substantial memory and latency overhead. For instance, generating a single video with MotionClone can take approximately seven minutes and 30 GB GPU memory.
>
> To address these challenges, we leverage the critical insight that each training video inherently serves as an ideal reference video for its corresponding training prompt. Additionally, we design a separate preprocessing phase to precompute the motion guidance before the actual training phase. As a result, this preprocessing phase eliminates the need for computationally expensive gradient calculations during training.
>
> As demonstrated in Tables 1 and 4, augmenting the teacher ODE solver with motion guidance leads to significant performance gains and improved motion quality across different evaluation metrics.
>
> > Additionally, while it conducts extensive ablation experiments on motion guidance and reward models, this does not constitute a significant contribution of the paper. I am unclear about the paper's contributions; is it providing more interesting insights? It would be helpful if the authors could briefly summarize this in their response.
>
> We firmly believe that our findings provide invaluable insights for advancing post-training research in video generation models. The choices of training data and reward models are arguably the most critical components in the post-training phase of a generative AI model. In our work, we conduct a rigorous and thorough empirical investigation into how these factors impact the performance of T2V models. A key finding of our study is that their effects are **not orthogonal**—the interaction between training datasets and RMs plays a pivotal role in shaping the final performance. Specifically, our Section 4.2, 4.3, and Appendix C.2 empirically demonstrate that **curating training datasets for different learning objectives is crucial for achieving optimal results**. To the best of our knowledge, this is the first work to systematically study how the selection of training data and RMs affects the post-training performance of a T2V model.
>
> > Regarding the contribution summary of the paper (L117), it seems to emphasize the advantages of existing work and the potential of extracting motion priors. And, I do not see any strong insights that stand out; motionclone has already demonstrated this fairly clearly. If the authors mean that motion priors are particularly useful during T2V training, they should provide more experiments. For example, training SVD and VideoCrafter2 shows that the insights presented in T2V-Turbo are quite limited.
>
> In our paper, we perform rigorous and thorough ablation studies to investigate the impacts of two critical components—training data and reward models—and demonstrate how different design choices affect model performance during the post-training phase. Our findings reveal the surprising **coupled effects of training data and RMs**, offering **strong insights** that can guide future research on post-training strategies for video generation models.
>
> Additionally, MotionClone only demonstrated its effectiveness during inference. Our work scales its application to the training phase by addressing two significant challenges: (1) its reliance on reference videos, and (2) the high computational cost of gradient calculations. Notably, our method is not confined to MotionClone’s motion guidance. It supports the integration of other energy functions into the teacher ODE solver, highlighting the vast design space for conditional guidance strategies. In this work, we use MotionClone’s energy function as a concrete example to showcase the potential of this approach, but its utility extends far beyond this specific application.

---

> > ### Author Response · Authors · 2024-11-21
> > **(2/2) Response to Reviewer dKgW**
> >
> > > I’m curious why this method, although based on distilling VideoCrafter2, outperforms VideoCrafter2 across multiple tasks, such as in the three metrics shown in Table 1.
> >
> > First, the performance differences can be attributed to the quality of the training datasets. As shown in Table 2, when trained exclusively on the lower-quality WV dataset, VCM (81.31/55.51/76.15) distilled from VideoCrafter2 (82.20/73.42/80.44) does not outperform VideoCrafter2. In contrast, training on higher-quality datasets, such as OV and VG, results in significant improvements, particularly in terms of visual quality. The key reason for this improvement lies in the nature of consistency distillation, which benefits greatly from high-quality training data. Higher-quality data produces better distillation targets, as outlined in Equation 4, enabling $\boldsymbol{f}_\theta$ to better regress toward latents that correspond to videos of higher quality.
> >
> > Second, our method incorporates additional feedback from a mixture of reward models trained to reflect human preferences, alongside learning from the motion priors extracted from the training videos. These additional supervision signals significantly enhance performance, as demonstrated in Tables 3 and 4.

---

> ### Author Response · Authors · 2024-11-24
> **Looking forward to your response!**
>
> Dear Reviewer dkgW,
>
> We greatly appreciate your insightful feedback, which has significantly contributed to the clarity and enhancement of our work. We have carefully addressed your comments in our response, clarified potential misunderstandings, and explained the technical contributions of our paper. Additionally, we also included new experimental results to corroborate our findings.
>
> We kindly invite you to revisit our paper in light of these updates and clarifications. We would greatly appreciate it if you could consider whether these changes warrant a reevaluation of your rating.
>
> Best regards,
>
> The Authors

---

> > ### Comment · Reviewer_dKgW · 2024-11-26
> >
> > Thank you to the authors for their response, which addressed most of my concerns. I have increased my score to 6.

---

> > > ### Author Response · Authors · 2024-11-26
> > > **Thank you!**
> > >
> > > Dear Reviewer dKgW,
> > >
> > > Thank you so much for recognizing our updates!
> > >
> > > Best regards,
> > >
> > > The Authors

---

### Official Review · Reviewer_y5Jq · 2024-11-03

**Soundness:** 4
**Presentation:** 3
**Contribution:** 3
**Rating:** 6
**Confidence:** 3

**Summary:**

This paper introduces T2V-Turbo-V2, aiming at improving the video quality and alignment with prompts by focusing the post-training phase. It distills a pre-trained T2V model using various supervision signals, including, 1. the reward models feedback from pre-trained vision language models (CLIP and InternV2) for both image and video levels. 2. the self-consistency loss used in many other distillation models, and injecting the classifier-free guidance and energy function into the self-consistency loss.
This paper also optimize the data preprocessing and the reward feedback, allowing the T2V-Turbo-V2 to achieve the state-of-the-art results on the Bench and outperform previous models like VideoCrafter2, Gen-3, and Kling.

**Strengths:**

1. The model achieves high scores across multiple metrics and outperforms proprietary systems, demonstrating the effectiveness of proposed modules.
2. T2V-Turbo-V2 introduce an effective post-training process to enhance video quality and prompt alignment, and this process if architecture agnostic, potentially can be used on other pre-trained video generation models.
3. This paper provide detailed ablation studies on various factors like the dataset selection, reward model configurations, the effectiveness of motion guidance, and so on.

**Weaknesses:**

1. The model appears optimized for single-caption, short-context prompts, its ability to generate longer or more complex video context may be limited.
2. When generating the dataset for the motion guidance, it may require considerable computation resources.

**Questions:**

1. What are the evaluation datasets when comparing the T2V-Turbo-V2 with the SOTA methods (Table 1)?
2. Since both the OV and VG dataset contain the high visual quality data, and the Quality Score is high when only using OV, it seems that OV is a good dataset to improve the visual quality, I want to know  the metrics of using OV + VG and OV + VG + WV (in Table 2 setting) and why the author does not use the OV dataset?
3. For the motion guidance, how the values of  λ, τ were chosen?
4. Can the author give more details on the dataset processing, like the needed computation resource?
5. If replace the base pre-trained video generation model with other models, can the T2V-Turbo-V2 method still achieve good results?

---

> ### Author Response · Authors · 2024-11-21
> **(1/2) Response to y5Jq**
>
> We thank the reviewer for their positive feedback on our work. Please find our detailed responses below.
>
> > The model appears optimized for single-caption, short-context prompts, its ability to generate longer or more complex video context may be limited.
>
> We would like to clarify the reviewer's misunderstanding. The limitation in generating longer or more complex video contexts stems from the text encoder used by the teacher diffusion model, not from our method. In this paper, we adopt VideoCrafter2 as our teacher model, which employs the CLIP text encoder with a context length of 77. This constraint is inherent to VideoCrafter2.
>
> However, this limitation can be easily overcome by using a more advanced teacher model, such as CogVideoX-5B or Mochi-1, both of which utilize the T5-XXL text encoder, supporting longer context lengths. Therefore, the limitation noted by the reviewer reflects a restriction of the teacher model (VideoCrafter2) rather than our proposed approach.
>
> > When generating the dataset for the motion guidance, it may require considerable computation resources
>
> We acknowledge that generating the dataset for motion guidance requires additional computational resources. Our preprocessing takes about 400 GPU hours, which is indeed manageable as it can be completed in approximately 2 days on a server with 8 NVIDIA A100 GPU (each GPU is of 40 GB memory). Importantly, this additional preprocessing effort yields significant benefits in terms of motion quality and inference acceleration.
>
> In comparison, the original MotionClone approach requires approximately 6 minutes to generate a single video, with 3 minutes spent on DDIM inversion from a reference video. Moreover, the peak GPU memory consumption can reach up to 35 GB due to the gradient calculations involved. In contrast, our model eliminates the need for identifying appropriate reference videos, conducting DDIM inversion, or calculating gradients during inference without adding any inference overhead compared to inference from a VCM. For example, our T2V-Turbo-v2 only takes 5 seconds to generate a video in 16 steps using BFloat16.
>
> The efficiency and performance gains of our approach are clearly reflected in its results. Our model achieves the #1 ranking on VBench, surpassing numerous proprietary video generation systems, including Gen-3, Kling, and MiniMax. This demonstrates that the preprocessing trade-off is well-justified, offering both superior motion quality and significant reductions in inference time.
>
> > What are the evaluation datasets when comparing the T2V-Turbo-V2 with the SOTA methods (Table 1)?
>
> The results in Table 1 are obtained by comparing our T2V-Turbo-V2 with the SOTA methods using the VBench datasets, which contain 946 unique prompts. We carefully follow VBench’s evaluation protocols by generating 5 videos for each prompt, as discussed in Lines 301 - 310.
>
> > I  want to know the metrics of using OV + VG and OV + VG + WV (in Table 2 setting)
> >
>
> We thank the reviewer for the question. Below, we include the results for OV + VG and OV + VG + WV in Table 2 setting.
>
> |  | **VCM (OV + VG)** | **VCM (OV + VG + WV)** | **T2V-Turbo-v2 w/o MG (OV + VG)** | **T2V-Turbo-v2 w/o MG (OV + VG + WV)** |
> | --- | --- | --- | --- | --- |
> | **Quality Score** | 83.43 | 82.95 | 83.86 | 82.35 |
> | **Semantic Score** | 57.52 | 54.98 | 69.25 | 76.80 |
> | **Total Score** | 78.25 | 77.36 | 80.94 | 81.24 |
>
> As shown in the table, VCM achieves a high Quality Score on the OV + VG dataset, similar to training on pure OV data, but adding the lower-quality WV data slightly decreases this score. Conversely, incorporating reward feedback in T2V-Turbo-v2 significantly improves the Semantic Score for the OV + VG + WV dataset, while the gains for OV + VG remain comparatively moderate. These findings align with our discussion in the main text: RMs with short context lengths operate optimally on datasets with shorter captions, highlighting the importance of aligning dataset characteristics with RM capabilities. Furthermore, the results justify our decision to exclude OV data when training the main T2V-Turbo-v2 models, as incorporating OV into VG + WV datasets negatively impacts model performance.
>
> > why the author does not use the OV dataset?
>
> We make our decision based on the performance when training on OV + WV. Specifically, VCM's Total Score when training on OV + WV (73.30) is worse than when training on OV (78.52) or WV (76.15). Similarly, our method's Total Score when training on OV + WV (81.00) barely improves the performance of training OV (80.97) and is worse than when training on WV (81.34) data. This phenomenon suggests a big domain gap between the OV and WV datasets. Thus, we do not use the OV dataset to train the main version of our method.

---

> > ### Author Response · Authors · 2024-11-21
> > **(2/2) Response to y5Jq**
> >
> > > For the motion guidance, how the values of λ, τ were chosen
> >
> > We choose λ = 0.5 based on the settings of MotionClone, which applies motion guidance for the first half of the inference calculation. Our value of τ = 500 is also based on MotionClone's choice and we found that slightly lowering its original value from τ = 2000 to τ = 500 leads to better performance and better training stability.
> >
> > > Can the author give more details on the dataset processing, like the needed computation resource?
> >
> > We spend 400 GPU hours for the data preprocessing phase, which can be completed with approximately 2 days on a server with 8 NVIDIA A100 GPU (each GPU is of 40 GB memory).
> >
> > > If replace the base pre-trained video generation model with other models, can the T2V-Turbo-V2 method still achieve good results?
> >
> > The original T2V-Turbo paper has demonstrated its performance when using both VideoCrafter2 and ModelScope as the teacher models. And we show that the technique of MotionClone can also be applied to ModelScope. And thus, we expect our T2V-Turbo-v2 can also achieve good results when using ModelScope as the base model.

---

> > > ### Comment · Reviewer_y5Jq · 2024-11-25
> > >
> > > We thank the authors time to add the quantitative results for my questions, and the explanation of other concerns, like the computation resource, the hyper-parameter choice, etc. I will keep my initial rating.

---

> > > > ### Author Response · Authors · 2024-11-25
> > > > **Thank you!**
> > > >
> > > > Dear Reviewer y5Jq,
> > > >
> > > > Thank you for your responses! If our responses have addressed your concerns, could you kindly increase the confidence score to better advocate the acceptance of our work?
> > > >
> > > > Thanks and best regards,
> > > >
> > > > The Authors

---

> ### Author Response · Authors · 2024-11-24
> **Follow-up the discussion**
>
> Dear Reviewer y5Jq,
>
> We greatly appreciate your insightful feedback, which has significantly contributed to the clarity and enhancement of our work. We have carefully addressed your comments in our response, clarified potential misunderstandings, and included the results for OV + VG and OV + VG + WV in the same settings as Table 2 to explain why we do not use the OV datasets to train our main models.
>
> We kindly invite you to revisit our paper in light of these updates and clarifications. We hope our responses address your concerns thoroughly and provide additional support for advocating the acceptance of our paper, potentially leading to an improved rating.
>
> Best regards,
>
> The Authors

---

### Official Review · Reviewer_N6o1 · 2024-11-04

**Soundness:** 3
**Presentation:** 3
**Contribution:** 2
**Rating:** 6
**Confidence:** 4

**Summary:**

This paper introduces T2V-Turbo-v2, an improved text-to-video model that enhances video generation by distilling a consistency model from a pretrained T2V model during the post-training phase. The method integrates multiple supervision signals—including high-quality training data, reward model feedback, and conditional guidance—into the consistency distillation process. Experiments show a new state-of-the-art result on VBench.

**Strengths:**

1. The paper is well-organized and easy to follow.
2. The method employing motion guidance is logically sound, and the experimental results showing improved semantic scores effectively validate its effectiveness.

**Weaknesses:**

1. The early work Energy-guided stochastic differential equations[1] first present a framework that utilize an energy function to guide the generaion process for diffusion model. Please cite this paper.
2.	In Figure 2, does the DDIM inversion require k forward passes for each training step? If so, does this introduce excessive computational cost?
3.	Please provide .mp4 files for visual comparisons, as Vbench cannot fully substitute for a user study. Including video files will allow reviewers and readers to better assess the performance and quality of the proposed method.

[1] The method employing motion guidance is logically sound,

**Questions:**

See weakness.

---

> ### Author Response · Authors · 2024-11-21
> **Response to Reviewer N6o1**
>
> > The early work Energy-guided stochastic differential equations[1] first presents a framework that utilizes an energy function to guide the generation process for a diffusion model. Please cite this paper.
>
> Can you please include the citation for the paper you mentioned? We are happy to cite the paper in our manuscript.
>
> > In Figure 2, does the DDIM inversion require k forward passes for each training step? If so, does this introduce excessive computational cost?
>
> As our paper already mentions, the inverse DDIM used to obtain motion prior can be done in a separate preprocessing phase and thus will **NOT** introduce any additional computational cost for training.
>
> > Please provide .mp4 files for visual comparisons
>
> Thank you for the suggestions. We have included the original video files in the supplemental material. On the other hand, you can click to play the videos in our manuscript if you open it using Adobe Acrobat Reader.

---

> ### Author Response · Authors · 2024-11-24
> **Follow-up the discussion**
>
> Dear Reviewer N6o1,
>
> We greatly appreciate your time and feedback on our work. We have carefully addressed your comments and clarified potential misunderstandings. Additionally, we also included new experimental results to corroborate our findings.
>
> We kindly invite you to revisit our paper in light of these updates and clarifications. We would greatly appreciate it if you could consider whether our responses warrant a reevaluation of your rating.
>
> Moreover, please take time to include the citation for the paper you mentioned. We are happy to cite the paper in our manuscript.
>
> Best regards,
>
> The Authors

---

> ### Author Response · Authors · 2024-11-26
> **Looking forward to your response!**
>
> Dear Reviewer N6o1,
>
> Thank you again for serving as a reviewer! Your feedback has been valuable in helping us clarify and improve our work. We have worked diligently to address your comments and included experimental results.
>
> We would like to invite you to continue the discussion with us. We hope that our responses can successfully address your concerns so that you might consider a reevaluation of your rating.
>
> Thanks and best regards,
>
> The Authors

---

> > ### Comment · Reviewer_N6o1 · 2024-11-26
> >
> > Dear Authors,
> >
> > Thank you for your detailed response and for addressing my concerns. The additional ablation studies on the data are particularly interesting and valuable. Based on the clarifications and supplementary experiments, I have increased my score.
> >
> > I do, however, have some additional questions and points of curiosity:
> >
> > 1. If one were to distill larger-scale diffusion models, such as CogVideoX, but lacked access to corresponding large-scale training datasets, do you think the existing open-domain academic datasets would suffice for effective distillation? From your experiments, did you observe any noticeable decline in visual quality based on human evaluation?
> >
> > 2. Could you elaborate on the differences and unique challenges between video distillation and image distillation? For instance, can existing image distillation techniques be directly applied to video models? From my understanding, many successful video diffusion models fundamentally extend image diffusion by treating videos as higher-dimensional imagess, without changing the underlying modeling approach.
> >
> > Lastly, I’d like to apologize for missing the related work reference in my initial review, which is titled EGSDE: Unpaired Image-to-Image Translation via Energy-Guided Stochastic Differential Equations.
> >
> > Thank you again for your efforts.

---

> > > ### Author Response · Authors · 2024-11-26
> > > **Response to the further questions**
> > >
> > > Dear Reviewer N6o1,
> > >
> > > We sincerely appreciate that you increased the rating for our work! We have cited the EGSDE paper in our revised manuscript.
> > >
> > > Additionally, please find our response to your questions below:
> > >
> > > > Do you think the existing open-domain academic datasets would suffice for effectively distilling larger-scale diffusion models, such as CogVideoX?
> > >
> > > Thank you for the great question! As demonstrated in our paper, consistency distillation can be performed efficiently. In Table 2, we show that VCM distilled from VideoCrafter2 achieves a superior Quality Score compared to its teacher model by training on OV or VG datasets for just 8,000 gradient steps. Our training was conducted on a machine with 8 GPUs and a total batch size of 24 (3 * 8), equating to approximately 200,000 text-video pairs. Both OV and VG datasets are densely captioned, open-sourced, and readily accessible, making them viable options for distilling larger-scale diffusion models like CogVideoX. Notably, recent work [1] has successfully trained video generation models capable of generating long videos using entirely open-sourced data, including OV and WV datasets.
> > >
> > > [1] Jin et al. Pyramidal Flow Matching for Efficient Video Generative Modeling. arxiv: 2410.05954.
> > >
> > > > Did you observe any noticeable decline in visual quality based on human evaluation?
> > >
> > > VCM’s visual quality can surpass that of its teacher model when the number of sampling steps is increased. However, videos generated by the student VCM model may exhibit decreased text-video alignment. This observation motivated our incorporation of reward models to address and mitigate this performance drop.
> > >
> > > > Could you elaborate on the differences and unique challenges between video distillation and image distillation? For instance, can existing image distillation techniques be directly applied to video models?
> > >
> > > Thank you for the insightful question. Fundamentally, existing techniques for distilling image diffusion models can be directly applied to video diffusion models, as video tensors are essentially stacks of image tensors. However, video distillation introduces unique challenges, particularly in modeling cross-frame dependencies. For instance, it is crucial to evaluate whether image-based distillation methods might degrade motion quality when applied to video models. Our method provides a solution to address the potentail quality loss by incorporating reward objectives and conditional guidance as additional supervision signals.

---

### Official Review · Reviewer_PqeK · 2024-11-05

**Soundness:** 3
**Presentation:** 2
**Contribution:** 2
**Rating:** 6
**Confidence:** 4

**Summary:**

This paper presents T2V-Turbo-v2, a method that enhances a diffusion-based text-to-video model by distilling a consistency model using high-quality training data, reward model feedback, and conditional guidance. The approach achieves state-of-the-art performance on VBench, demonstrating improved text-video alignment and motion quality through tailored datasets, diverse reward models, and optimized guidance strategies

**Strengths:**

The results appear promising and solid.

The experiments are thorough.

The writing is easy to follow.

**Weaknesses:**

The method combines existing techniques such as consistency distillation and motion guidance, so its novelty is somewhat limited.

VideoCrafter uses a 2D+1D decoupled spatial-temporal approach, whereas most recent advanced methods employ full 3D attention. How would motion guidance be applied when using 3D attention?

What is the peak training cost, such as peak GPU memory, compared to training a single model? How does it perform on less powerful video diffusion models like Zeroscope—can it still achieve results comparable to VideoCrafter?

**Questions:**

Please see the weakness part.

---

> ### Author Response · Authors · 2024-11-21
> **Response to Reviewer PqeK**
>
> We thank the reviewer for their feedback on our work. Please find our detailed response below.
>
> > The method combines existing techniques such as consistency distillation and motion guidance, so its novelty is somewhat limited.
>
> We respectfully disagree with the reviewer's argument on our novelty. In terms of guidance strategy, the primary goal of our work is not to design a new energy function for motion guidance. Instead, we aim to empirically demonstrate that augmenting the teacher ODE solver with the energy function's gradients of a conditional guidance strategy can **distill a more capable student video generator while significantly reducing inference costs**. In this context, we showcase the potential of our approach by integrating MotionClone’s energy function into the teacher ODE solver.
>
> Additionally, adapting MotionClone’s motion guidance techniques for training is significantly non-trivial for several reasons:
>
> - **Dependency on Reference Videos**: MotionClone relies on access to reference videos with high-quality motion. However, identifying suitable reference videos for general text prompts is challenging, which limits its effectiveness and applicability for generic video generation tasks.
> - **High Computational Cost**: Computing the gradient of the energy function during inference incurs substantial memory and latency overhead. For instance, generating a single video with MotionClone can take approximately seven minutes and 30 GB GPU memory.
>
> To address these challenges, we leverage the critical insight that each training video inherently serves as an ideal reference video for its corresponding training prompt. Additionally, we design a separate preprocessing phase to precompute the motion guidance before the actual training phase. As a result, this preprocessing phase eliminates the need for computationally expensive gradient calculations during training.
>
> As demonstrated in Tables 1 and 4, augmenting the teacher ODE solver with motion guidance leads to significant performance gains and improved motion quality across different evaluation metrics.
>
> > VideoCrafter uses a 2D+1D decoupled spatial-temporal approach, whereas most recent advanced methods employ full 3D attention. How would motion guidance be applied when using 3D attention?
>
> We would like to address the reviewer’s understanding of our contributions. The applicability and generalizability of MotionClone’s technique are beyond the scope of our paper. The primary goal of our work is not to design a new energy function for motion guidance.
>
> Instead, we aim to empirically demonstrate that **augmenting the teacher ODE solver with the energy function gradients of a conditional guidance strategy can distill a more capable student video generator while significantly reducing inference costs**. To illustrate this, we integrate MotionClone’s energy function into the teacher ODE solver, highlighting the potential of this approach.
>
> Nonetheless, MotionClone’s motion guidance can also be applied to advanced methods, such as Open-Sora, Open-Sora-Plan, and Latte [1]. As shown in Figure 3 of [2], these models employ a similar DiT-based architecture featuring two types of Transformer blocks: spatial and temporal. Since the temporal Transformer blocks process information across temporal dimensions, MotionClone’s success can be replicated within these frameworks with minimal adaptation. For methods utilizing full 3D attention, e.g., CogVideoX, we can always reshape the attention matrix to obtain temporal attention across different video frames. And thus, full 3D attention should not be a barrier to leverage MotionClone’s techniques.
>
> [1] Ma et al., Latte: Latent Diffusion Transformer for Video Generation.
>
> [2] Zhao et al., Real-Time Video Generation with Pyramid Attention Broadcast.
>
> > What is the peak training cost, such as peak GPU memory, compared to training a single model?
>
> The peak training cost of our method is **identical** to that of training a single model. As highlighted in our paper, the motion prior is extracted during a separate preprocessing phase and does **not** contribute to the peak GPU memory usage during training.
>
> > does it perform on less powerful video diffusion models like Zeroscope—can it still achieve results comparable to VideoCrafter?
>
> T2V-Turbo’s success has already been demonstrated with two different teacher models: VideoCrafter2 and ModelScope. Regarding the generalizability of MotionClone, the original MotionClone paper conducted its experiments using AnimateDiff [3] as the base model, and our work successfully extended this technique to VideoCrafter2. Therefore, there is no inherent barrier to applying this approach to less powerful models like Zeroscope, and we expect it our T2V-Turbo-v2 achieve similar successful results when employing Zeroscope as the teacher model.

---

> ### Author Response · Authors · 2024-11-24
> **Looking forward to your response**
>
> Dear Reviewer PqeK,
>
> We greatly appreciate your time and feedback on our work. We have carefully addressed your comments and clarified potential misunderstandings. Additionally, we also included new experimental results to corroborate our findings.
>
> We kindly invite you to revisit our paper in light of these updates and clarifications. We would greatly appreciate it if you could consider whether our responses warrant a reevaluation of your rating.
>
> Best regards,
>
> The Authors

---

> > ### Author Response · Authors · 2024-11-26
> > **Follow up the discussion**
> >
> > Dear Reviewer PqeK,
> >
> > Thank you again for serving as a reviewer! Your feedback has been valuable in helping us clarify and improve our work. We have worked diligently to address your comments and included experimental results.
> >
> > We would like to invite you to continue the discussion with us. We hope that our responses can successfully address your concerns so that you might consider a reevaluation of your rating.
> >
> > Thanks and best regards,
> >
> > The Authors

---

> > > ### Author Response · Authors · 2024-11-27
> > > **Let's discuss!**
> > >
> > > Dear Reviewer PqeK,
> > >
> > > Thank you once again for your time and efforts in providing valuable feedback on our work. Your insights have been instrumental in helping us refine and improve our submission.
> > >
> > > We would like to kindly invite you to follow up on the discussion regarding our work. If you have any additional comments or concerns, please don’t hesitate to let us know, and we will do our utmost to address them promptly.
> > >
> > > The Authors

---

> > > > ### Author Response · Authors · 2024-11-28
> > > > **Happy Thanksgiving!**
> > > >
> > > > Dear Reviewer PqeK,
> > > >
> > > > On this Thanksgiving, we would like to take the opportunity to express our heartfelt gratitude for your time and effort in providing valuable feedback on our work. Your insights have been truly invaluable in helping us refine and improve our submission.
> > > >
> > > > We kindly invite you to follow up on the discussion regarding our work. Should you have any additional comments or concerns, please don’t hesitate to let us know—we are committed to addressing them to the best of our ability.
> > > >
> > > > The Authors

---

> > > > > ### Author Response · Authors · 2024-11-30
> > > > > **Thank you again!**
> > > > >
> > > > > Dear Reviewer PqeK,
> > > > >
> > > > > We hope you are having a wonderful weekend! Once again, we sincerely appreciate your valuable feedback on our work.
> > > > >
> > > > > We kindly invite you to follow up on the discussion, and we would be happy to address any additional concerns or questions you might have.
> > > > >
> > > > > The Authors

---

> > > > > > ### Comment · Reviewer_PqeK · 2024-11-30
> > > > > >
> > > > > > Thank you to the authors for their responses. After thoroughly reviewing the rebuttals and considering the concerns raised by other reviewers, I still hold the opinion that this method heavily depends on MotionClone and is challenging to apply to 3D full transformers (as simply modifying the shape of attention maps does not effectively decouple spatial and motion aspects). However, I am willing to increase my score to 6 and leave the final decision to the area chair.

---

> > > > > > > ### Author Response · Authors · 2024-12-01
> > > > > > > **Thank you!**
> > > > > > >
> > > > > > > Dear Reviewer PqeK,
> > > > > > >
> > > > > > > Thank you so much for raising your rating on our work!
> > > > > > >
> > > > > > > Again, we would like to emphasize that **the primary goal of our research is not to design a superior energy function to improve motion guidance**. Instead, we aim to highlight the vast design space of conditional strategies and demonstrate their potential to enable a more capable student video generator without adding inference overhead.
> > > > > > >
> > > > > > > In this paper, we empirically show that leveraging MotionClone's energy function enhances the motion quality of the generated videos. Thus, we believe our paper successfully achieves its proof-of-concept mission, **paving the way for future research to distill knowledge from a more diverse set of energy functions.**

---

> > > > > > > > ### Author Response · Authors · 2024-12-04
> > > > > > > > **MotionClone can indeed be applied to transformers with full 3D attention!**
> > > > > > > >
> > > > > > > > Dear Reviewer PqeK,
> > > > > > > >
> > > > > > > > Thank you again for your insightful comments. We would like to comment further on MotionClone's applicability to transformers with full 3D attention.
> > > > > > > >
> > > > > > > > As per MotionClone's authors' [comments](https://openreview.net/forum?id=aY3L65HgHJ&noteId=rxABkhmCT5), **MotionClone can indeed be applied to the latest DiT-based T2V model, e.g., CogVideoX**, in which MotionClone demonstrates effectiveness in training-free motion customization.
> > > > > > > >
> > > > > > > > We hope it can address your concern about MotionClone's applicability!
> > > > > > > >
> > > > > > > > Thanks and Best regards,
> > > > > > > >
> > > > > > > > The Authors

---

### Author Response · Authors · 2024-11-21
**General response: new experimental results!**

We appreciate the reviewers for their time and constructive feedback on our work. We have responded to individual reviews below and would like to clarify some common misunderstandings about our work.

1. We firmly believe that our rigorous and thorough ablation studies on the training data and reward models provide valuable insights for future research on post-training strategies for video generation models. Our findings reveal the surprising **coupled effects of training data and RMs**, highlighting the importance of **curating datasets tailored to specific learning objectives to achieve optimal results**. To the best of our knowledge, this is the first work to systematically examine how the selection of training data and RMs impacts the post-training performance of a T2V model.
2. **Our method is not confined to MotionClone’s motion guidance.** Instead, we leverage MotionClone’s energy functions to highlight the vast design space of conditional guidance strategies, which enables the distillation of a more capable student video generator while significantly reducing inference costs.
3. **Adapting MotionClone’s motion guidance techniques for training is significantly non-trivial** due to 1) its dependency on reference videos and 2) high computational costs. We address these challenges by leveraging the critical insight that each training video is an ideal reference video for its corresponding training prompt. Additionally, we design a separate preprocessing phase to precompute the gradients of energy functions, enabling efficient training.
4. **Trade-offs of the Preprocessing Phase**. While our preprocessing phase introduces additional computational overhead (~400 GPU hours), it yields significant improvements in motion quality and accelerates the inference process.

Additionally, we include two new experiment results:

1. In Appendix E, we conduct experiments on OV + VG and OV + VG + WV to **corroborate the results in Table 2** and clarify why we excluded OV data when training our main models.
2. In Appendix F, we conduct a **human evaluation** to compare the 16-step generation of T2V-Turbo-v2 w/o MG and T2V-Turbo-v2 w/ MG, confirming that incorporating motion guidance significantly enhances model performance and improves the motion quality of generated videos.

---

### Meta-Review · Area_Chair_FUJY · 2024-12-16

**Metareview:**

This paper presents T2V-Turbo-v2, a method that enhances a diffusion-based text-to-video model by distilling a consistency model using high-quality training data, reward model feedback, and conditional guidance. The overall writing is good and easy to follow.

Several reviewers raise questions about the limited novelty and more experiments, with mixed reviews. During the rebuttal and refined version, all the issues are solved, leading to borderline acceptance for all reviewers.

The area chair checks the rebuttal stage, questions, and responses, suggesting the acceptance of this work as a poster.

**Additional Comments On Reviewer Discussion:**

All the issues are well solved during the long discussion stage.

---

### Decision · Program_Chairs · 2025-01-22

Accept (Poster)